# A shallow deep learning approach to classify skin cancer using down-scaling method to minimize time and space complexity

**Sidratul Montaha**[1], **Sami Azam**[2]*, **A. K. M. Rakibul Haque Rafid**[1], **Sayma Islam**[1],
**Pronab Ghosh**[3], **Mirjam Jonkman**[2]

**1** Department of Computer Science and Engineering, Daffodil International University, Dhaka, Bangladesh,
**2** College of Engineering, IT and Environment, Charles Darwin University, Casuarina, NT, Australia,
**3** Department of Computer Science, Lakehead University, Thunder Bay, ON, Canada

* sami.azam@cdu.edu.au

**Data Availability Statement:** A total of 3297 dermoscopy images of skin cancer image ISIC archive are studied in this research which can be accessed from this link https://www.kaggle.com/

## Abstract

The complex feature characteristics and low contrast of cancer lesions, a high degree of inter-class resemblance between malignant and benign lesions, and the presence of various artifacts including hairs make automated melanoma recognition in dermoscopy images quite challenging. To date, various computer-aided solutions have been proposed to identify and classify skin cancer. In this paper, a deep learning model with a shallow architecture is proposed to classify the lesions into benign and malignant. To achieve effective training while limiting overfitting problems due to limited training data, image preprocessing and data augmentation processes are introduced. After this, the 'box blur' down-scaling method is employed, which adds efficiency to our study by reducing the overall training time and space complexity significantly. Our proposed shallow convolutional neural network (SCNN_12) model is trained and evaluated on the Kaggle skin cancer data ISIC archive which was augmented to 16485 images by implementing different augmentation techniques. The model was able to achieve an accuracy of 98.87% with optimizer Adam and a learning rate of 0.001. In this regard, parameter and hyper-parameters of the model are determined by performing ablation studies. To assert no occurrence of overfitting, experiments are carried out exploring k-fold cross-validation and different dataset split ratios. Furthermore, to affirm the robustness the model is evaluated on noisy data to examine the performance when the image quality gets corrupted.This research corroborates that effective training for medical image analysis, addressing training time and space complexity, is possible even with a light-weighted network using a limited amount of training data.

## 1. Introduction

Cancer is one of the most severe threats to global health in today's world. Globally, approximately 19.3 million new cancer cases, including 1,198,073 skin cancer cases, and nearly 10.0 million cancer deaths, including 63,731 skin cancer deaths, were recorded in 2020 [1]. The global cancer burden is estimated to rise by 47%, with an occurrence of 28.4 million new cases,

fanconic/skin-cancer-malignant-vs-benign. The dataset was divided into two folders of train and test. There are two classes, benign and malignant, where a total of 1800 dermascopy images are found benign and the rest 1497 images are malignant. All the images were in RGB format and have 224 x 224 pixels. However, all the rights of the Data are bound to the ISIC-Archive rights https://www.isic-archive.com. The ISIC Archive is an open source platform with publicly available images of skin lesions under Creative Commons licenses or CC BY-NC license. This type of license allows others to distribute, remix, adapt, and build upon the material in any medium or format for noncommercial purposes only, and only so long as attribution is given to the creator. Basically this license allows the creator to retain copyright whilst allowing others to copy and distribute and make use of their work non-commercially. This archive serves as a public resource of images for teaching, research, and for the development and testing of diagnostic artificial intelligence algorithms. Anyone can access ISIC archive and make use of the skin lesion images that it provides in their research work. ISIC-Archive contains a total of 69445 skin cancer dermoscopy images of two classes with necessary metadata. Interested researchers can go to the above link and access the images by clicking on 'Gallery' option. From this page, images can be downloaded with metadata by selecting required number of pictures. However, images of ISIC-Archive are in different pixel size. Therefore, if any researcher wants a small portion of labeled and resized dataset of ISIC-Archive, they might work with 'Skin Cancer: Malignant vs. Benign' dataset that we have employed. This is a labeled and resized dataset which is publicly visible and hence anyone can view and access the dataset. Further information regarding the access procedure of the data origin point can be found in this GitHub repository https://github.com/hrafid/ISIC-data-availability-statement/blob/5fa2ab56db6bc0f6cddac0b2df171841a9391bc6/Data%20availability%20statement.pdf.

**Funding:** The author(s) received no specific funding for this work.

**Competing interests:** The authors have declared that no competing interests exist.

**Abbreviations:** SCNN_12, shallow convolutional neural network; MSE, Mean Squared Error; SSIM, Structural similarity index measure; PSNR, peak signal-to-noise ratio; RMSE, Root Mean Squared Error; DSC, Dice Similarity Co-efficient; MAE, Mean absolute error; AUC, Area under curve; ROI, Regions of Interest; CNN, Convolutional neural network; R-CNN, Region Based Convolutional

by 2040 [1]. Around 75% of deaths related to skin cancer are caused by malignant melanomas. To lessen the spread of cancer and increase survival rates worldwide, early cancer screening and diagnosis are considered essential public health tactics. Skin cancer can be remedied if it is detected early when the thickness of the malignant tumor is lower than a particular threshold. However, the prognosis worsens rapidly as the disease progresses. The five-year survival rate of advanced stage cases is less than 15%, while the survival rate for early-stage diagnosis with proper treatment is above 95% [2]. Regrettably, since 2020, the diagnosis and treatment of cancer is been hindered due to the pandemic of coronavirus disease 2019 (COVID-19) as fear of infection with the virus in health care systems, inhibits people from seeking screening, diagnosis, and treatment for non–COVID-19 disorders [3]. For early skin cancer screening, a full body skin checkup, which is reasonably quick and non-invasive, is recommended by medical associations. Nonetheless, visual skin examination might not be an effective approach to discover skin cancer at an early stage [4]. Medical appliances may aid doctors in screening but due to the symptoms and characteristics of skin cancer, such as the potential of rapid progression requiring diagnostic refinement, accurate diagnosis of melanoma is difficult even for experts [4]. Melanoma, the worst form of skin cancer, is the least frequent yet the deadliest type of lesion and it can spread swiftly to other regions of the body. It develops when melanocytes undergo malignant transformation [5]. As the correspondence between malignant and benign features is particularly close at the initial evolutional phase [6], early detection remains challenging for clinical experts.

Dermoscopy is a non-invasive examination procedure where the diagnostic accuracy depends on the experience and training of the dermatologist. Over the past decades, dermoscopic examination has been employed progressively in clinical practice to assess vascular structures [7]. Unfortunately, even using dermoscopy, the accuracy of melanoma detection is estimated to be only 75–84% [8]. Though it improves the accuracy of diagnosis significantly, assessment of dermoscopic images is time-consuming and errors may occur during the process even with expert dermatologists [9]. The morphological structural features of a lesion, such as the color spectrum, shades of color, skin lesion color gradient, number of lesions, streaks, and grid patterns, are the major causes behind this [10]. Moreover, people often only consult doctors after cancer has progressed to the point that it is impossible to recuperate from. In most cases, the initial observation is done in primary care clinics before the patient is transferred to a dermatologist as it is quite unachievable for dermatologists to check all patients for primary staged skin tumors [11]. In this case, computer aided diagnosis can play a significant role as it processes by extracting key details such as color variant, texture features, asymmetry, border irregularity, and diameter which may not be perceived by human eyes accurately [12]. Image processing techniques along with machine learning methods could be a useful approach for classification and diagnosis but often cause false positive and false negative cases. Deep Learning technology could aid to detect skin cancer in an early stage with higher accuracy and reduce process interpretation time.

The aim of this research is to classify skin cancer lesions into benign and malignant classes, employing a deep learning approach. A shallow deep learning model is proposed by conducting an ablation study on it. As dermoscopic images are affected by noise, hairs, dark corners, color charts, uneven illumination and marker ink [13], image pre-processing methods are deployed to increase the performance of our proposed model. Several commonly used algorithms are introduced to segment the skin lesions accurately. Moreover, as deep learning requires a dataset of sufficient size, data augmentation is carried out to increase the number of images. To reduce training time and space complexity, a down-scaling approach is introduced using bilateral and box blur algorithm. In our study, the term down-scaling means, reducing total space taken by an image without decreasing its actual size. Simply, the image size will

Neural Networks; FrCN, Full resolution convolutional networks; BCC, Basal cell carcinoma; ANFC, Adaptive Neuro-Fuzzy Classifier; VGG-16, Visual Geometry Group-16; SGD, stochastic gradient descent; CLAHE, Contrast limited adaptive histogram equalization; SURF, Speeded-Up Robust Features; CAD, Computer Aided; FC, fully connected layers; APIs, Application Programming Interfaces; GPU, Graphical Processing Units; TP, true positive; TN, true negative; FP, False positive; FN, False negative; Acc, Accuracy; F1, F1-score; TPR, true positive rate; FPR, False positive rate; ROC, Receiver operating characteristic curve; T_loss, Training loss; T_acc, training accuracy; V_loss, validation loss; V_acc, validation accuracy; Te_loss, test loss; Te_acc, test accuracy; OP, optimizer; LR, learning rate; Rec, Recall; Spe, Specificity; Pre, Precision; BC, benign; MC, malignant; SGDM, Stochastic Gradient Descent with momentum.

remain same, but the space taken by image will be reduced. After applying the method, statistical analysis, including Mean Squared Error (MSE), Peak Signal-to-Noise Ratio (PSNR), Root Mean Squared Error (RMSE), Structural similarity index measure (SSIM), Dice Similarity Coefficient (DSC) and histogram analysis, is done on the processed image to evaluate the algorithms. As a result of introducing down-scaling method, the image size and model training time, denoting space complexity and time complexity, are reduced significantly. Performance evaluation matrices such as Accuracy (ACC), Recall and Specificity, F1-score, Root mean squared error (RMSE), Mean absolute error (MAE) and Area under curve (AUC) value are computed to evaluate the effectiveness of our model. However, to check the possibility of occurring overfitting, k-fold cross-validation using different dataset split ratios are performed on both datasets before and after applying augmentation. Moreover, as real world dataset contains noises, the model is evaluated on noisy data to examine the performance when the image quality gets corrupted.

This paper is organized in a total of 11 sections. Firstly, the challenges of skin cancer detection are demonstrated in section 2 and research objectives are shown in section 3. A detailed literature review with limitations is presented in section 4. Dataset used in this study is described in section 5. Section 6 contains some sub-sections that describe the image preprocessing steps, image down-scaling method and image segmentation process. Later, augmentation of down-scaled images and overview of the training approaches is presented in section 7. In section 8, the proposed model SCNN_12 is described. Section 9 showcases experimental results and discussion along with all the findings of this study. Limitations of this research is presented in section 10 and finally the conclusion of the study is presented in section 11

## 2. Challenges in skin cancer detection

Challenges for detecting skin cancer in an early stage can be described as follows:

1. Noise may obscure significant features of an image, and artifacts may interfere with obtaining the desired accuracy. Therefore, noise and unwanted regions which may contain artifacts should be eliminated.

2. In some cases, there is low contrast and brightness from neighboring tissues, color texture, light rays and reflections form additional impediments to scrutinizing skin lesions accurately.

3. Moles of the human body, which can be quite similar to benign and malignant lesions, make classification hard.

4. The structure of cancerous lesion of benign and malignant stages is nearly similar for which it is often challenging to interpret correctly without proper image preprocessing steps.

   Fig 1 shows several challenges commonly included in skin cancer dermoscopy images.

## 3. Research objectives

The main objective of this study is to construct an effective method to classify skin lesions into benign and malignant classes with the highest possible accuracy while limiting time and space complexity. The main contributions of our paper are as follows:

1. Unnecessary regions of tissue and hairs visible in images are removed using morphological closing.

2. The quality of the images is improved by applying the enhancement technique, gamma correction.

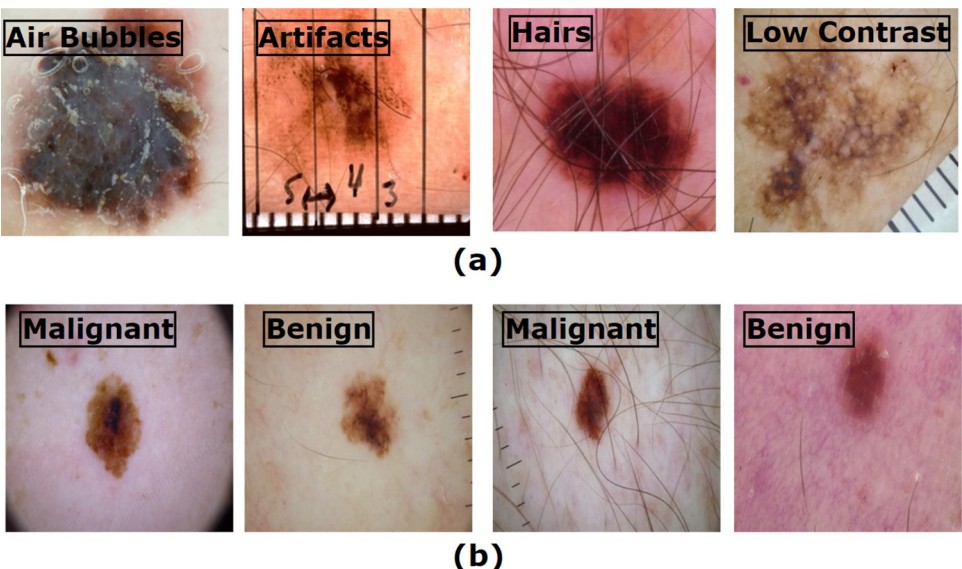

**Fig 1. Automatic recognition of skin cancer is impeded by various factors.** (a) represents some interferences with important features (b) shows the similarity between benign and malignant lesions.

3. Regions of Interest (ROI) are extracted successfully using different algorithms, namely OTSU thresholding, morphological dilation, subtraction and morphological opening.

4. Down-scaling is carried out by employing the box blur algorithm followed by a bilateral filter to boost the performance while reducing computational complexity.

5. The number of dermoscopy images is increased by employing four color space data augmentation techniques in order to reduce overfitting issues while training the model.

6. A shallow convolutional neural network (SCNN_12) model of depth 12 is proposed, after performing an ablation study, to classify skin cancer with the highest possible accuracy.

7. The model is tested using K-fold cross-validation and splitting the dataset using various ratios before and after augmentation to check overfitting issue.

8. The robustness of the model is ascertained, evaluating the model on noise induced test dataset.

## 4. Literature review

To improve the efficiency of the performance of automatic detection, segmentation and classification, many specialists and scholars have conducted research in this area. Ameri A. et al., [14], proposed a deep CNN model to classify skin cancer images into two classes. AlexNet was employed as the pre-trained model. A total 3400 images, 1700 in the benign and 1700 in the malignant class, were obtained from the HAM10000 dermoscopy image database. Their proposed model obtained an accuracy of approximately 84%. The existing public skin cancer datasets mostly contain an inadequate number of segmented ground truth labeling which is laborious and expensive. However, on their dataset, almost 80% of images were at benign class. Authors manually balanced the classes with the equal number of images that led to higher accuracy. In this case, implementing some effective image processing and data augmentation techniques could have been a better approach. Moreover, the proposed model might be

further experimented by changing different parameters and hyper-parameters that might aid to improve robustness and a good comparison among the performance could have been presented. As segmentation of cancerous lesions accurately is vital, the authors of a recent paper [15], proposed fully automated deep learning ensemble methods, based on Mask R-CNN and DeeplabV3+, for the segmentation of lesion boundaries. For pre-processing the datasets, the authors applied the Shades of Gray algorithm. They used the ISIC-2017 segmentation training set for training their network. The model performed its best by achieving the highest accuracy of 94% using Fuzzy Gradient Descent (FGD) optimizer. The ISIC-2017 testing set and PH2 dataset were used as the test set to evaluate the performance of their introduced methods. For the ISIC-2017 testing set, their proposed ensemble methods performed preeminently in segmenting the skin lesions with a sensitivity of 89.93% and a specificity of 97.94%. In terms of sensitivity, the suggested Ensemble-A approach beat FrCN, FCNs, U-Net, and SegNet by 4.4%, 8.8%, 22.7%, and 9.8%, respectively. Their proposed Ensemble-S, Ensemble-L and Ensemble-A networks performed best with an accuracy of 93%, 93% and 94% respectively. In this study, exploring more data pre-processing and augmentation techniques might had an impact on increasing accuracy as skin cancer dataset often contains noises, artifacts and limited number of images. Moreover, an ablation study could have been experimented tofigure out how the model behaves under different parameters. Kharazmi et al [16], developed a novel automatic skin vessel segmentation framework in order to detect and evaluate cutaneous vascular structures in ceroscopy images. The retrieved vascular characteristics were further investigated for skin cancer categorization into Basal cell carcinoma (BCC) and benign lesions. Applying their framework, a computer-assisted disease classification was performed to differentiate BCC from benign lesions. The authors obtained a segmentation sensitivity and specificity of 90% and 86% respectively on a test set of 500000 manually segmented pixels, defined by an expert as the ground truth. Compared to some other state of the art methods, the proposed method achieved the highest AUC of 96.5% when separating BCC from benign lesions using only the extracted vascular features and the Random Forest classifier. One of the limitations of this study is that they did not eliminate the bubble presented in skin images. As the presence of artifacts highly interfere on overall performance, removal of artifacts might have been a better approach to improve their accuracy. Later, a new approach, based on a novel regularizer technique was introduced to evaluate the classification accuracy of a deep CNN model [17]. The regularizer was based on the standard deviation of the weight matrix of the classifier and embedded in each convolution layer to control the values of kernel matrix. The suggested model obtained an accuracy of 97.49% and an AUC value of 98.3% for 100 epochs outperforming other advanced techniques. However, the authors admitted that the proposed regularizer cannot be utilized for feature selection or reduction. They also mentioned choosing optimal gamma value for gamma correction algorithm is computationally expensive and time intensive. Sikkander et al. [18] proposed a novel segmentation based classification model for the identification of skin lesions on ISIC skin lesion dataset by merging a Grab-Cut algorithm with an Adaptive Neuro-Fuzzy Classifier (ANFC). The system was developed by deploying four key steps namely, preparation, segmentation, extraction of features, and classification. After preprocessing the images by applying the top hat filter and inpainting technique, the GrabCut algorithm was executed to extract the ROI. Afterwards, the feature extraction phase was carried out by using a deep learning based Inception model. Finally, a dynamic hybrid ANFC system was employed to classify the dermoscopy images. A dataset with seven classes obtained from the ISIC skin lesion dataset was used for this purpose. The proposed method performed with an accuracy of 97.91%, a sensitivity of 93.40% and a specificity of 98.70%. Their future work includes, experimentation with other deep learning networks that might improve overall accuracy. Jinnai et al. [19] trained a faster, region-based

CNN (FRCNN) for the classification of benign and malignant tumors on a dataset of 4732 clinical images. The model was trained for 100 epochs using Visual Geometry Group-16 (VGG-16) as the backbone network. To improve the performance, data augmentation was carried out by applying some commonly used techniques. However, a maximum accuracy of only 91.5% was achieved by evaluating the model on a test dataset of 666 images. Other authors [20] attempted to increase the proficiency of deep CNNs using transfer learning and fine-tuning for the classification of benign vs. malignant skin cancer. They used a dataset of 3600 images of size 224 × 224, downloaded from the ISIC Archive. Initially, three pretrained models namely Inception v3, InceptionResNetv2 and ResNet50 were employed on ISIC dataset of 3600 images for 20 epochs with a learning rate of 0.0001, Adam optimizer and a batch size of 32. These models were then trained and evaluated by changing the optimizer and batch size to find the best performance. The highest experimental accuracy, precision, recall, F1 score and ROC-AUC score of 93.5%, 94%, 77%, 85% and 86.1% were obtained with ResNet50 using the Stochastic Gradient Descent (SGD) optimizer and a batch size of 64. Although, they performed data augmentation, no image processing method was carried out in this study, which is one of the crucial steps in skin cancer classification tasks. Along with batch size and optimizer, learning rate might be changed since, learning rate highly impact on performance and training time. After extracting the ROIs from the images of DermIS and DermQuest datasets using an improved k-mean algorithm, a new fully automated CNN based transfer learning approach was used for the identification of melanomas [21]. To address overfitting issues, data augmentation was applied on the ROIs of the images. They proposed a fine-tuned model based on AlexNet architecture, by replacing the low level feature layers resulting in the highest accuracy of 97.9% and 97.4% on DermIS and DermQuest datasets respectively. However, in pre-processing step, only a noise removal technique is applied. Artifacts removal, image enhancement and experimenting with some optimizer and learning rate while training the models, could have been incorporated to validate the model's robustness. Artifacts and resemblances between the normal and cancerous skin lesions can reduce the performance of a model [22]. Therefore, authors of this paper, removed these artifacts in order to implement the segmentation process more precisely. They subsequently applied YOLOv4 deep neural network and active contour segmentation method in order to segment the melanoma tumor. The model was tested on the ISIC2018 and ISIC2016 datasets and achieved an average dice score of 1 and a Jaccard coefficient of 98.9%. Wei et al [23] proposed a proficient and shallow melanoma classification network based on MobileNet, DenseNet architecture, with a small number of parameters. A lightweight U-Net model was built to segment skin lesion regions. The quantity of the images was improved from 900 to 4430 by utilizing different augmentation methods. Their model achieved an accuracy of 96.2%. Input image size of their network was too large that required higher computing resources. Due to insufficient amount of data, in some cases poor performance was observed. Besides, experimenting some loss functions could have been incorporated to enhance model's performance. The entire literature review including limitation is summarized in Table 1 as shown below.

## 5. Dataset description

A total of 3297 dermoscopy images, provided by the Kaggle skin cancer image ISIC archive [24] dataset are analyzed for this research. There are two classes, benign and malignant, where 1800 of the dermascopy images are benign and 1497 are malignant. The images are found in RGB format and have 224 x 224 pixels. The description of the dataset is summarized in Table 2 and Fig 2.

**Table 1. A summary of the literature review showing the past techniques and their limitations.**

| Authors | Task | Models | Limitations |
|---|---|---|---|
| Ameri et. al., [14] | Classification, segmentation | CNN | i. Further experimentation of proposed model's parameters is absent. <br> ii. Absence of image pre-processing and data augmentation technique |
| Manu Goyal et. al., [15] | segmentation | Mask R-CNN DeeplabV3+ | i. Lack of ablation study on proposed model <br> ii. Lack of image processing and data augmentation techniques that might have given better accuracy |
| Kharazmi et. al., [16] | Classification | Feature extraction Random forest | i. No eliminatation of artefacts (bubble) that are present in the images <br> ii. Absence of ablation study in proposed model |
| Albahar et. al., [17] | Classification | CNN model with novel regularizer | i. Lack of ablation study in proposed model |
| Sikkander et. al.. [18] | Segmentation Classification | ANFC | i. Experimentations with other deep learning models is absent. |
| Sagar et.al., [20] | Classification | CNN | i. Absence of image preprocessing techniques <br> ii. Use of a specific optimizer and learning rate |
| Ashraf et. al., [21] | Segmentation | YOLOv4 | i. Lack of artefacts removal techniques <br> ii. Use of a specific optimizer and learning rate |
| Wei et. al., [23] | Classification | CNN | i. Size of input image is too large requiring higher resources <br> ii. Experimentations with various loss functions is absent |

## 6. Proposed methodology

This section constitutes three main sub-sections: image preprocessing, segmentation and down-scaling. Algorithms used for these processes are described briefly in this section.

In image preprocessing, under the sub-sections hair removal and enhancement, two algorithms namely morphological closing and gamma correction are applied respectively. The preprocessed resulting image is then passed through the steps of segmentation and down-scaling. Six techniques related to segmentation are introduced, namely Otsu's thresholding, morphological dilation, subtraction, morphological erosion, largest blob and hole-filling to extract the ROI successfully. Down-scaling is carried out on preprocessed images by employing two algorithms: bilateral filter and box blur, where bilateral filter is used to smooth the pixels and box blur is used to down-scale the images. A number of assessment techniques, namely, MSE, PSNR, SSIM, RMSE, Histogram Analysis and DSC are applied to the down-scaled images to compare the structural and feature similarity measures.

### 6.1. Image pre-processing

Image preprocessing involves suppressing undesired features of an image as well as enhancing meaningful features such as color, shape and texture which are necessary for a particular application. The raw image dataset which is downloaded from a source has often unwanted regions, small air bubbles, blood vessels, variations in brightness, contrast and noise as well as hair presented in various densities [25]. This could result in a reduced performance of the proposed neural network based model. As shown in Fig 3, a malignant melanoma image can be

**Table 2. Dataset description.**

| Name | Description |
|---|---|
| Total number of images | 3297 |
| Dimension | 224 × 224 pixels |
| Color grading | RGB |
| Benign | 1800 |
| Malignant | 1497 |

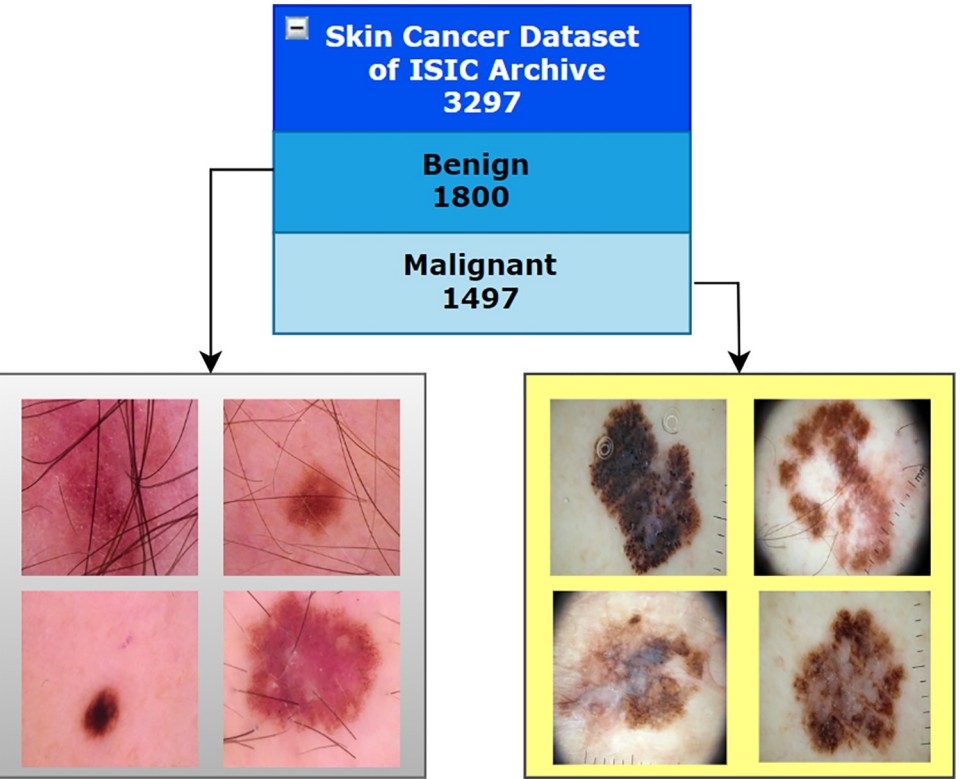

**Fig 2. Skin cancer ISIC archive dataset.**

visualized as three channels: healthy skin, cancerous lesions and artifacts. Artifacts, including unwanted objects and hair are denoted as peripheral and the cancerous lesion is denoted as the tumor. These regions are also characterized by variations in size and multi-scale and multi-resolution features [26]. Therefore, for each image, we focused on the elimination of artifacts surrounding the tumor and the extraction of the cancerous lesion. Several methods (e.g. median filter, morphological closing, k-means clustering) have been introduced in the literature to automate the preprocessing and lesion segmentation in this context.

These algorithms are applied to our images and the best one, based on highest PSNR values is selected as PSNR is a prominent statistical measure to compare the original and processed images. A higher PSNR value indicates a higher quality and a less noisy image. Algorithms applied for image processing and their PSNR values, are summarized in Table 3.

Here, three processes are mentioned namely, hair removal, contrast enhancement and noise removal. For hair removal, two algorithms are picked and since morphological closing yields the highest PSNR score, closing is employed for the process. For contrast enhancement, between contrast limited adaptive histogram equalization (CLAHE) and gamma correction, gamma correction is chosen as the PSNR value of gamma correction is higher than CLAHE. Likewise, in the last process, instead of selecting median filter, bilateral filter is chosen for noise removal and down-scaling image pixels.

Methodologies applied to meet the challenges in skin cancer detection are illustrated in (Fig 4). Here, the complete preprocessing method is divided into two main processes: 'hair removal' and 'enhancement'. In this approach, the tiny blood vessels, air bubbles and hairs are eliminated through the morphological closing operation which is a sub-process of 'hair removal'. In the 'enhancement' process, gamma correction is applied to enhance the contrast.

**Fig 3. Three regions and artifacts in the skin lesion image.**

**6.1.1. Hair Removal.**   Since the presence of hairs can reduce the classification accuracy and overall performance of a model, they should be eliminated before the training phase. For this purpose, morphological closing is adopted as it can successfully eliminate the hairs without destroying the meaningful features.

*6.1.1.1 Morphological Closing.* Morphological closing is a variant form of a morphological operation that consists of morphological dilation followed by erosion. In this regard, the

**Table 3. Comparison of different approaches based on PSNR values.**

| Process | Algorithm | PSNR |
|---------|-----------|------|
| Hair removal | Morphological tophat + Impainting | 39.23 |
| | Morphological Closing | 40.54 |
| Contrast Enhancement | CLAHE | 38.22 |
| | Gamma correction | 41.29 |
| Noise remove / Down-scaling | Median filter | 41.78 |
| | Bilateral filter | 42.94 |

dilation operation fills narrow gaps and notches of the contour and extends the thickness or size of the foreground object according to the shape of a given structuring element or kernel while the erosion operation shrinks the foreground object of the resultant image with the same kernel [27]. In this process, a kernel is structured of a suitable size based on the operation to be performed. Fig 5 illustrates the process flow of morphological closing on our dataset.

The implementation is fairly straight forward and is accomplished by using morphologyEx (), a function of openCV. The function will perform dilation and erosion and return the desired output. The mathematical expression of morphological closing of X by B, as explained by [28] can be stated as Eq 1:

$$X \cdot B = (X \oplus B) \ominus B \tag{1}$$

where, X = input image, B = structuring element, X∘B is closing of X by B, $X \oplus B$ = dilation of X by B, $(X \oplus B) \ominus B$ = erosion of $X \oplus B$ by B, $\oplus$ = symbol of addition and $\ominus$ = symbol of subtraction. Therefore, the closing of X by B = the dilation of input image, X followed by the erosion.

The formula of dilation of X by B can be stated as Eq 2:

$$X \oplus B = \bigcup_{b \in B} X_b = \bigcup_{x \in X} B_x = \{x + b \mid x \in X, b \in B\} \tag{2}$$

here, the kernel B is positioned with its origin at (x, y) and the new pixel value is derived using the formula of Eq 3 [29]:

$$g(x \cdot y) = \begin{cases} 1 \text{ if } F \text{fits } S \\ 0 \text{ otherwise} \end{cases} \tag{3}$$

here, F denotes the image where erosion will be performed and S denotes the structuring element for performing erosion. The formula of erosion of X by B, where $\mathcal{E}$ is an integer grid and X is an image in $\mathcal{E}$ can be stated as Eq 4:

$$X \ominus B = \bigcap_{b \in B} X_{-b} = \{p \in \mathcal{E} \mid B_p \subseteq X\} \tag{4}$$

here, $B_p$ indicates translation of B by p and translation of X by -b is denoted by $X_{-b}$.

This function cv2.morphologyEx() is based on the concept of bitwise filtering which requires three parameters namely input image, type of morphological operation and

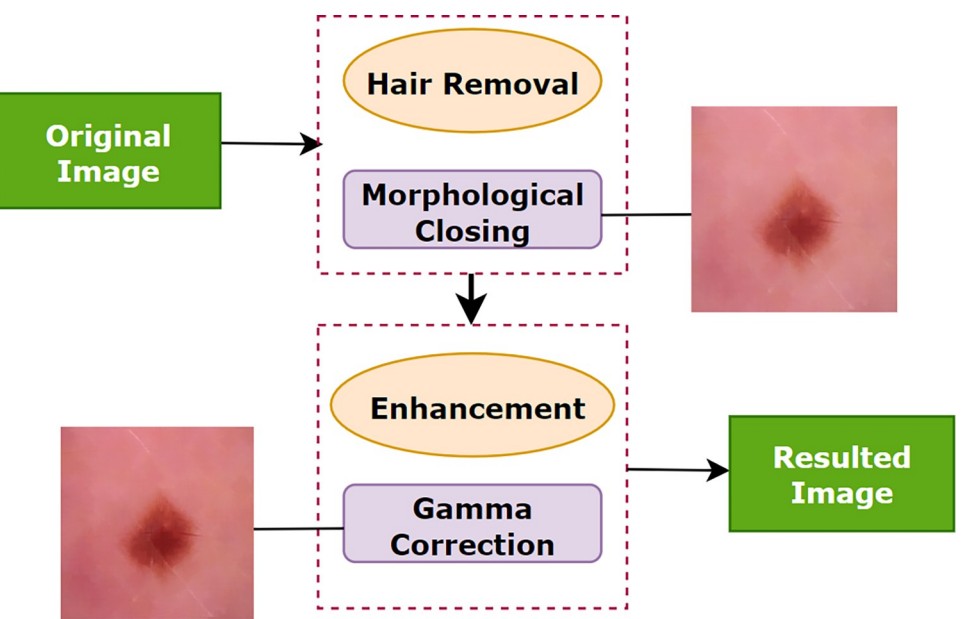

**Fig 4. Steps performed for image pre-processing.**

structuring element or kernel which determines the nature of the operation. For our case, the type of morphological operation is morphological closing (cv2.MORPH_CLOSE). Before applying the filter, a structuring element also known as kernel is generated. Using NumPy, a kernel can be structured in a rectangular shape. But a rectangular kernel does not always produce better result, especially in complex structured medical images. In these special cases, an elliptical or circular shaped mask may provide a better outcome. With the help of cv2.getStructuringElement () which is a function of opencv, a structuring element of different shapes can be generated. This function requires two parameters: kernel shape and size.

Kernel shape: the kernel shape can be rectangular, elliptical or circular. The shape is chosen based on the characteristics of the images and the goal of applying the kernel.

Kernel size: kernel size is also set based on the application. If a very tiny noise is present in an image, kernel size can be smaller and vice versa.

In our study, we have used a Cross-shaped kernel as it provided the optimal output while not degrading the image quality. However, rectangular and elliptical shaped kernel are also experimented which did not provide a satisfactory output as shown in Fig 6. Although hairs

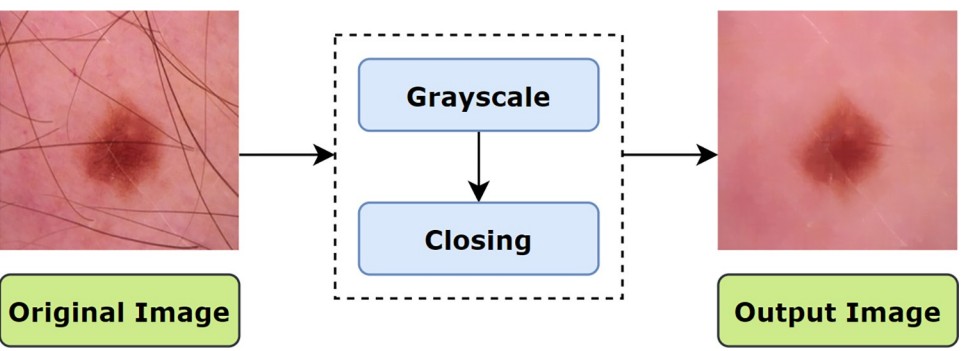

**Fig 5. Process flow of applying morphological closing.**

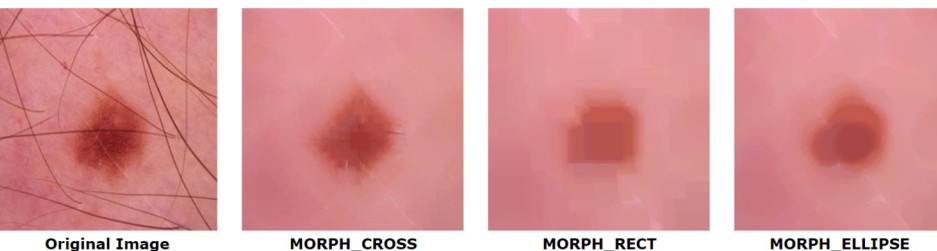

| Original Image | MORPH_CROSS | MORPH_RECT | MORPH_ELLIPSE |

**Fig 6. Resultant image of morphological closing with different kernel shape.**

were removed, image quality decreased and became highly pixelated. Our kernel size for structuring element was (15, 15). As the hair was spread all over the image, a small kernel size (e.g. 5, 5) could not remove all the hairs successfully.

The result is that hairs, both inside and surrounding the foreground object, are successfully removed from the input image. The process flow of morphological closing and the resultant image are shown in Fig 6.

**6.1.2. Enhancement.** The main goal of image preprocessing is image restoration or enhancement in order to enhance the relevant details by applying a suitable algorithm, based on the nature of the dataset. In order to recognize objects, an image needs to be processed with illumination compensation [12]. Since the gamma correction algorithm performed with the highest PSNR score for this purpose, we have introduced the method for this study.

*6.1.2.1. Gamma Correction.* Gamma correction, also regarded as 'Power Law Transform', carries out a non-linear transformation to every pixel of a source image [29]. Instead of adding a constant value to the pixels, gamma correction applies an exponential function on individual pixel intensity values. This can be expressed mathematically as Eq 5:

$$g(x, y) = [f(x, y)]^{\gamma} \tag{5}$$

where, $g(x,y)$ is the gamma corrected image, $f(x,y)$ is the input image and $\gamma$ is the given gamma value. A gamma value, $\gamma < 1$ is considered as encoding gamma causing the lesion to be brighter in a dark background and a gamma value $\gamma > 1$ is considered as decoding that causes the lesion to be darker in a light background. Encoding gamma restores the pixels more efficiently. The process of applying gamma correction can be illustrated as Algorithm 1.

```
Algorithm 1. Gamma Correction.
BEGIN
    (1) Read Input image, f(x,y) where and x and y represents the
dimensions
    (2) Apply gamma value, γ on f(x,y)
    (3) Let, Output image = g(x,y)
    (4) IF γ = 1:
        Output, g(x,y) = linear.
    (5) ELSE IF γ < 1:
      Output, g(x,y) = brighter
    (6) ELSE IF γ >1:
      Output, g(x,y) = darker
END
```

The gamma distribution are plotted in 3 different graphs as shown in Fig 7, illustrating the output behavior of an image for different gamma values. In each of the graphs, the x-axis represents the intensity values of the input image and the y-axis represents the intensity values of the resultant image. A significant difference is noticed in the input and output intensity with changes of the gamma value.

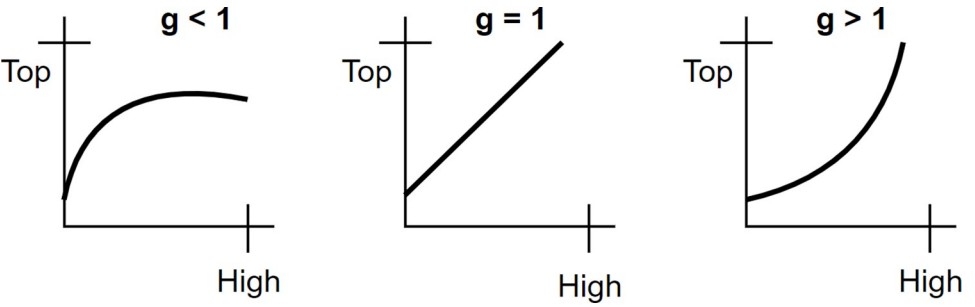

**Fig 7. Plots for three gamma correction states.**

A suitable gamma can be found by experimenting with different gamma values on the target image [30]. In this process, the first step requires rescaling the pixel intensity range from [0,255] to [0, 1.0]. After selecting and applying a suitable gamma value based on the color intensity of the source image, the target output is obtained. Finally, the resulting image is scaled back again to [0,255]. In our study, a gamma value of 1.2 is used, as higher gamma values were washing out the pixels and lower gamma values cause a loss of important details. Fig 8 illustrates image transformation for different gamma values. For a gamma value 1, the output is linear. For gamma values of 0.1 and 0.5, the image is found quite faded. Moreover, for the gamma valuesof 1.5 to 3.0, the image is found to become darkened. In both cases, cancerous lesion is not clear and less highlighted which might cause poor performance of the neural network.

However, a universal value of 1.2 would not be suitable for any image sets in which skin cancer lesions should be detected but a suitable gamma value can be chosen following our process.

The process flow of applying gamma correction and changes in the resultant image is shown in Fig 9 where the output of morphological closing is used as the input of gamma correction.

Table 4 represents all the determined parameter values for morphological closing and gamma correction which are described above.

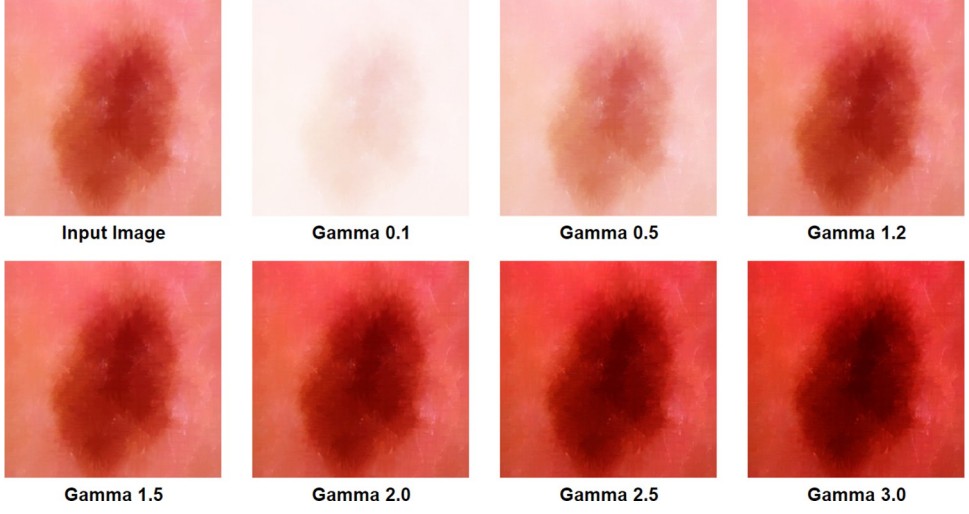

**Fig 8. Gamma corrected image transformation.**

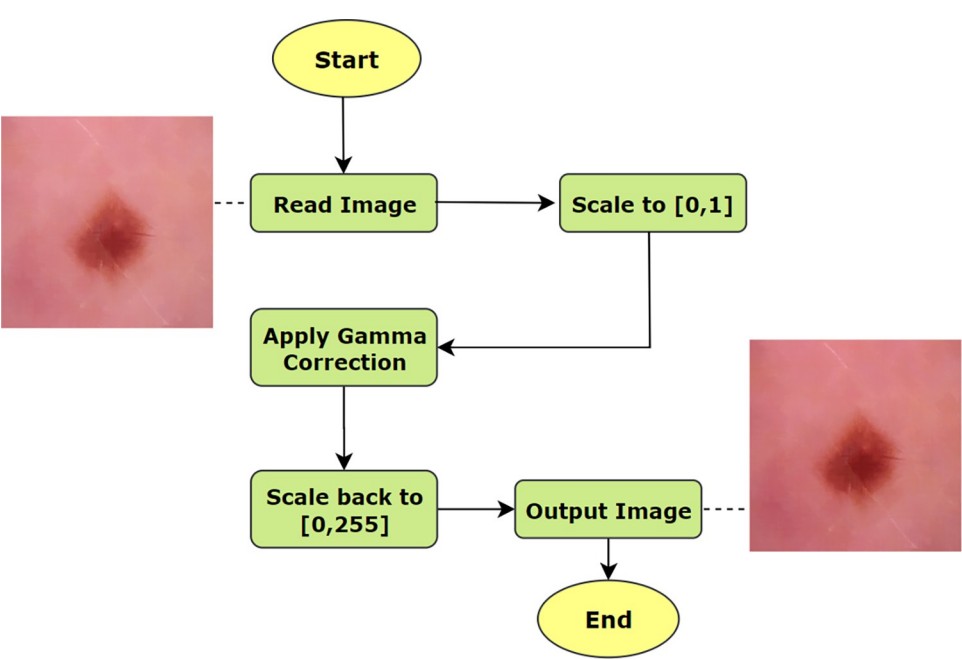

**Fig 9. Process flow of applying gamma correction.**

## 6.2. Segmentation

Segmentation of medical images by extracting the disease area is a crucial component in this field of research [31]. The detection of the ROI is a complex task and segmentation of melanocytic lesion can be very difficult [32]. For this purpose, several techniques are employed to bring out the desired lesion from the images. The data enhancement method used in this segmentation task is the same as described in section 6.1. In the segmentation step, the Otsu's thresholding, morphological dilation, substraction, morphological erosion, largest contour detection and hole-filling methods are carried out to extract the cancerous lesion from the surrounding healthy skin. The process of image segmentation is shown in Algorithm 2.

```
Algorithm 2. Image segmentation process.
BEGIN
    1. Read preprocessed dermoscopy images, p(x,y) where and x and y
represents the dimensions
    2. Apply Otsu's thresholding on p(x,y)
      Let otsu_img, o(x,y) is the output.
    3. Apply morphological dilation on otsu_img, o(x,y)
      Let dilate_img, d(x,y) is the output
    4. Subtract otsu_img from dilate_img, d(x,y)
      Let sub_img, s(x,y) is the output
    5. Apply morphological erosion on sub_img, s(x,y)
      Let erote_img, e(x,y) is the output
    6. Find the largest blob from erote_img, e(x,y)
      Let blob_img, b(x,y) is the output
      Apply hole-filling operation on blob_img, b(x,y)
      Let hole_img, h(x,y) is the output
    7. Apply inrange operation on hole_img, h(x,y) and preprocessed
image p(x,y)
      Let segmented_img, f(x,y) is the final output
END
```

**Table 4. Selected parameter values for morphological closing and gamma correction.**

| algorithm | parameter |
|---|---|
| Morphological closing | Structuring element = cv2.MORPH_CROSS, kernelSize = (15, 15) <br> morphological operation = cv2.MORPH_CLOSE |
| Gamma correction | Gamma value = 1.2 |

**6.2.1. Otsu's Threshholding.** A key prerequisite for extracting the essential features is that the lesion must be distinct from the surrounding normal skin. To achieve that, automatic thresholding proposed by Otsu is one of the most powerful methods [12]. This is a non-linear operation that converts a gray-scale image into a binary image. The input of this algorithm is generally a grayscale image and while the output is a binary image based on the pixel intensity of the input image. If the intensity of a pixel > threshold, the corresponding output pixel is marked as 1 (white), and if the input pixel intensity < = threshold, the output pixel is marked as 0 (black). The equation for deriving the threshold value can be illustrated as Eq 6:

$$T = \frac{1}{2}(\mu_1 + \mu_2) \tag{6}$$

For computing a threshold value automatically between 0 and 1, Otsu's thresholding follows the basic procedure of Algorithm 3 [13]:

**Algorithm 3. Otsu's thresholding.**
**BEGIN**
 1. Read an image as input
 2. Select an initial estimated threshold value for T.
 3. Using the estimation of T, segment the image which produces two groups of pixels: G1 and G2. Therefore, G1 consisting of all pixels with the intensity values ≥T, and G2 consisting of all pixels with the intensity values <T.
 4. Calculate the mean intensity values µ1 and µ2, for the pixels in both regions G1 and G2.
 5. Derive a new threshold value based on the formula of Eq (3).
 6. Repeat steps 2 through 4 until the difference in T< the predefined parameter T0.
**END**

In this method, the algorithm establishes an optimal threshold value by minimizing the within-class variance ($\sigma^2$) or maximizing the between class variance ($\sigma^2$) of an object and background pixels of an image. The between-class variance formula can be defined as [13]:

$$\sigma^2(t) = w_0(t)w_1(t)[\mu_1(t) - \mu_2(t)]^2 \tag{7}$$

$$\sigma^2(t) = w_0(t)\left[-w_0(t)\left[\frac{\mu - \mu_1(t)}{1 - w_0(t)} - \frac{\mu(t)}{w_0(t)}\right]\right] \tag{8}$$

$$w_0(t) = \sum_N^n w_1(t) = 1 - w_0(t), \mu(t) = \sum t\frac{n}{N} \tag{9}$$

where, σ2 (t) is the between-class variance, w is the weight, µ is the combined average value and T is the threshold value. In this process, pixels of the image are divided into two clusters with intensity values 0 and 255. Here, $w_0$, $\mu_1$ respectively represents background pixel weight and mean values and on the other hand, $w_1$, $\mu_2$ respectively represents foreground pixel weight

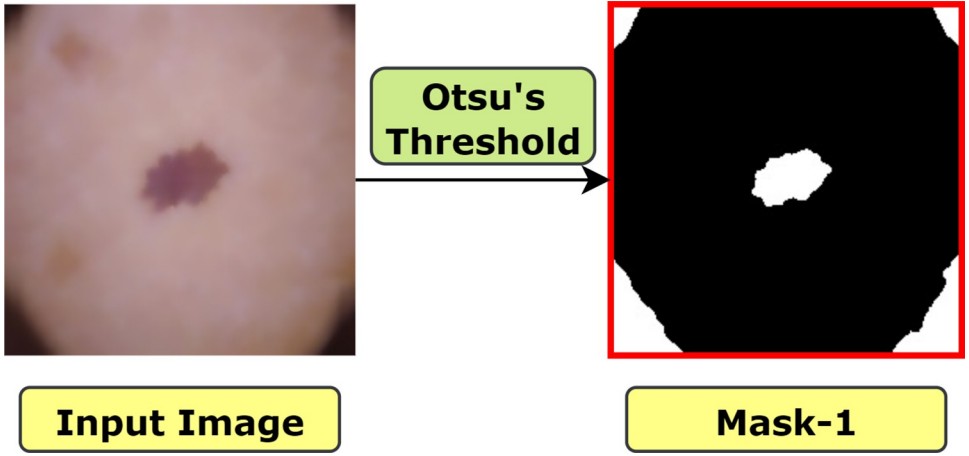

**Fig 10. Resultant image after applying Otsu's thresholding.**

and mean values. For our case, the algorithm is applied on skin images using cv2.threshold() method that requires four parameters:

Input image: a gray scale image where the algorithm will be applied. Our input image is skin cancer image after applying morphological closing and gamma correction.

Threshold value: This value is chosen based on the input pixel intensity. If the input pixel value is less than the threshold, it is set to 0, otherwise, a maximum value (usually 255) is used. Our threshold value was 120, as for skin lesion the pixel intensity was around this value in most of the images. However, this value is determined based on optimal resultant image, after experimenting with some other threshold values.

Maximum value: This value is assigned to the pixel that surpass the mentioned threshold. Usually, this value is 255 which is applied on our study as well.

Thresholding technique: As we have applied Otsu's thresholding, cv2.THRESH_OTSU is passed as an extra flag along with cv2.THRESH_BINARY. Therefore, our parameter is cv2. THRESH_BINARY+cv2.THRESH_OTSU.

The resultant image of Otsu's thresholding is shown in Fig 10. Here, the output image is denoted as 'mask-1' with a red border surrounding it, in order to show the white regions surrounding the four corners of the image more clearly.

**6.2.2. Morphological Dilation.** Dilation, represented by $\oplus$ is a base morphological operation that adds pixels to the boundaries of foreground objects. Valleys and the width of maximum regions are expanded using dilation by eliminating undesirable noises. cv2.dilate() method is used to apply morphological dilation which requires two inputs to perform the process. The first input is the original input image and the second input is the structuring element by which the image will be dilated. This structuring element, also known as the kernel, determines how many pixels will be added in order to emphasize the size of the foreground object [5]. In this case, our inputs are binarized image and a structuring element (kernel) of size (5, 5). We chose this kernel size as our goal is to enlarge the skin foreground lesion in a minimum amount. A higher kernel size would much expand the area which was not our purpose. The mathematical formula of morphological dilation is defined in section **6.1.1.1**.

The kernel size can be decreased and increased to get the desired shape and size of the foreground object, as illustrated in Fig 11. The blue border around the red object indicated how the size becomes larger after the dilation operation.

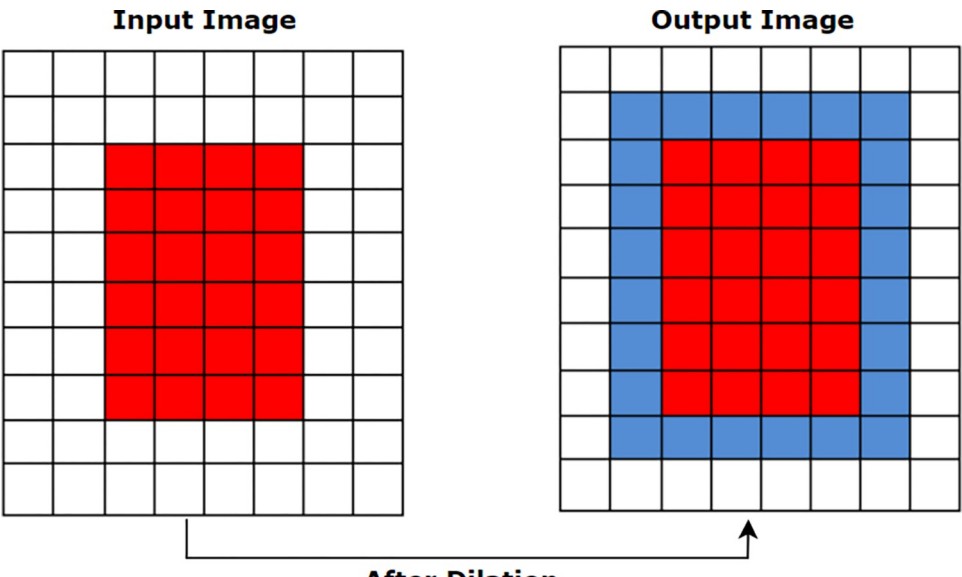

**Fig 11. Changes of pixels after applying morphological dilation.**

The resultant image (mask-2), after applying dilation on mask-1 is shown in Fig 12 where regions of mask-2 are expanded compared to mask-1.

**6.2.3. Subtraction.** In the previous operations (Otsu's thresholding and morphological dilation), two masks: mask-1 and mask-2 were generated where mask-2 was obtained by dilating the pixels of mask-1. The purpose of dilating the white region of mask-1 and producing mask-2 is to acquire the edge of the cancerous lesion. A simple subtraction of mask-1 from mask-2 is implemented to preserve the edges of all the contours presented in the image [27]. Thus, a new mask, mask3 is created, which is shown in Fig 13. The process is described in Algorithm 4. In Fig 13, the lesion of mask-2 is denoted as L-2 and the lesion of mask1 as L-1 where both mask-1 and mask-2 are of the same size of 224 X 224. Both lesions have pixel value of 255 where the number of pixels in L-2 > number of pixels in L-1. Therefore, while subtracting mask-1 from mask-2, all the overlapping white pixels of mask-2 and mask-1 result in black pixels. Algorithm 4 shows the pseudo-code of deriving mask-3:

```
Algorithm 4. Pixel subtraction method.
START
    READ mask-2 as input m2(x,y) and mask-1 as m1(x,y)
    CALCULATE N = len (m2)
    LET, m3(x,y) = output
    FOR pixel in N:
        IF m2[pixel] = = 255 AND m1[pixel] = = 255
        ASSIGN m3[pixel] = 0
    ELSE IF m2[pixel] = = 255 AND m1[pixel] = = 0
        ASSIGN m3[pixel] = 255
    ELSE IF m2[pixel] = = 0 AND m1[pixel] = = 0
        ASSIGN m3[pixel] = 0
    END IF
    END FOR
END
```

As L-2 and the other white regions (four corners) of mask-2 are greater than the white regions of mask-1, some white pixels remain after subtraction. As a result, thin white edges appear on mask-3 which is nothing but the dilated pixels of mask-2 (Fig 13).

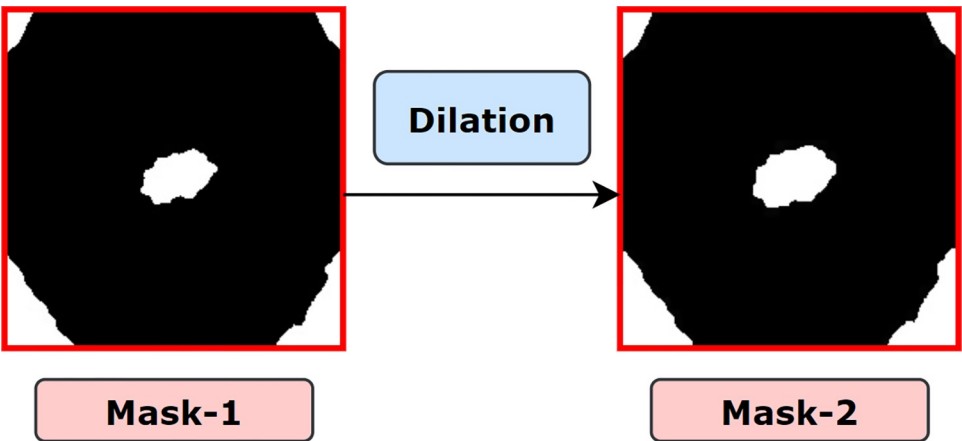

**Fig 12. Resultant mask after applying morphological dilation.**

**6.2.4. Largest blob.** It is observed from mask-3 that, along with the cancerous lesion, unnecessary regions surrounding the ROI have also been extracted. To eliminate these regions and keep only the desired ROI, the largest blob is detected [12]. Using the functions of OpenCV namely findContours () and drawContours (), it is possible to locate and acquire the size of contours in an image. A contour can be defined as a sequence of pixels that denotes a

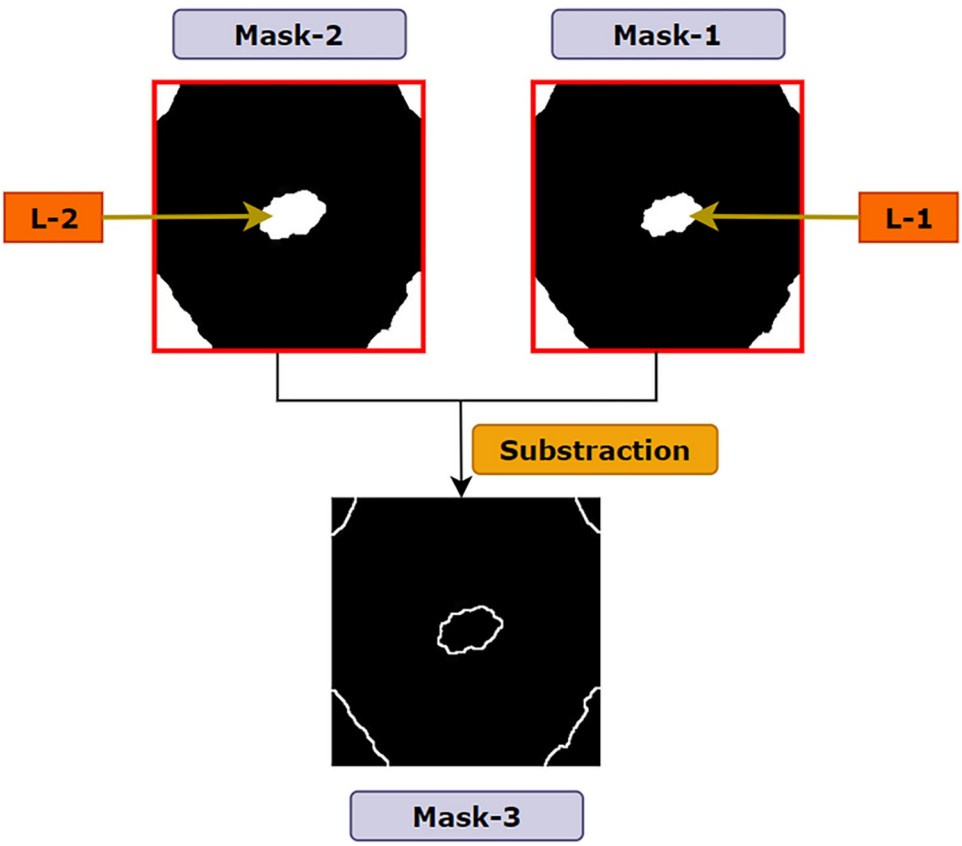

**Fig 13. Resultant mask after subtracting mask-1 from mask-2.**

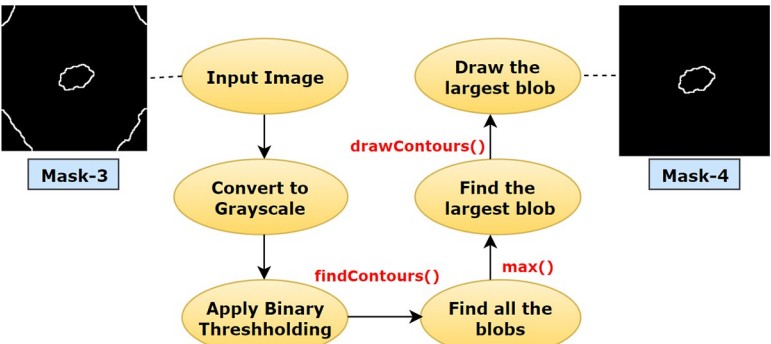

**Fig 14. Process flow of extracting the largest blob from mask-3.**

particular object in an image. in this process, at first mask-3 is converted into single channel grayscale format and binary thresholding is applied. Hence, all the objects in the image will be divided into two clusters, black (0) and white (255) based on the intensity value. All white pixels separated by borders of black pixels will be considered as a contour. With the help of the findContours (), all the objects in mask-3 are located. The function requires three parameters namely input image, contour retrieval mode and contour approximation method. It then provides a python list of all contours and a hierarchy as outputs. Here, our first parameter value is the skin cancer image (Mask-3 of Fig 14). Second parameter, contouring retrieval mode is cv2. RETR_EXTERNAL, since our goal is to extract the largest contour. This mode derives only the outermost contours presented in an image. We have used cv2.CHAIN_APPROX_SIMPLE as the third parameter value. Instead of storing all the coordinates of a contour, cv2.CHAIN_AP-PROX_SIMPLE preserves only the necessary coordinates. For example, for the contour of a rectangular shape, it does not retrieve all the points. Rather it just stores the necessary four points, removing the redundant points, by which a rectangle can be easily derived. We have used the approximation method as it saves memory while not compromising efficiency. Subsequently, after getting the list of contours, the function max () is employed to find the largest blob by sorting the list from largest to smallest using the attribute, reverse = True. We passed two parameters, contour list and a key = cv2.contourArea. Using the key, the function max() derives the largest blob from the contour list. Finally, the function drawContours() proceeds to draw an outline of this largest object area. After this process is completed, a binary mask is returned covering the desired largest blob [5].

The process flow of deriving mask-4 from mask-3 using the largest blob detection is illustrated in Fig 14.

**6.2.5. Hole-Filling.** After extracting the largest contour from the image, it can be noticed that the interior of the blob has intensity values similar to the background pixels. Therefore, the hole has to be filled to obtain the final mask for segmentation. The objective of filling a hole for a binary image is filling the entire region of a contour with the pixel value of 255 using Eq 10 [13].

$$X_k = (X_{k-1} \otimes B) \cap A^c \tag{10}$$

here, B is the structuring element or kernel, k defines the pixel index of an image where k denotes the pixel which is being altered and k-1 indicates the neighboring pixels of k. A is the

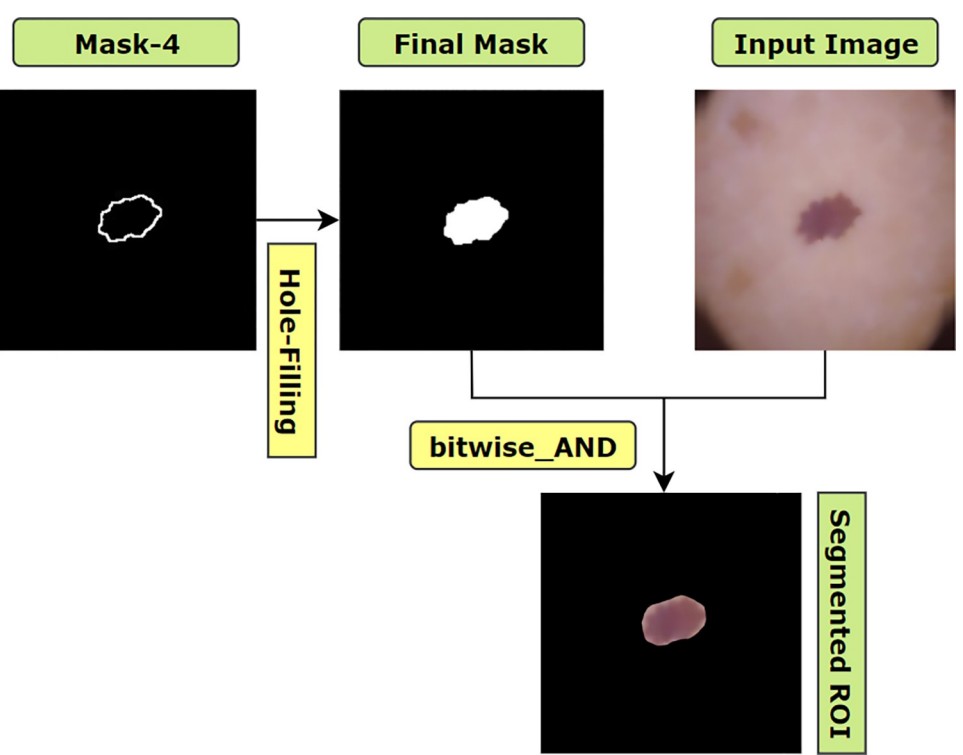

**Fig 15. Process of extracting ROI by filling the largest blob.**

set containing a subset according to the definition of [29], For hole filling, Eq 11 is ued.

$$F(x, y) = \begin{cases} 1 - I(x \cdot y) & \text{if } (x, y) \text{ is on the border of } I \\ 0 & \text{otherwise} \end{cases} \qquad (11)$$

where, I is the binary image and F is the output image. F will be 0 everywhere apart from the image border, which is denoted as 1−I. Finally, to draw the largest contour, cv2.drawContours function is utilized. Here, the first parameter is where the contour will be drawn; the second parameter is the list of contours. As we have just one largest contour, we have passed a list of single contour. The third parameter is the index of target contour to be drawn, for our case '0' is set, as we have just one index in the contour list. The fourth parameter is the color of marking. For our case, the value is (255,255,255). Therefore, the contour will be drawn with a pixel value of (255,255,255). The last parameter is the thickness of drawing the contour border. We have filled the contour area with a pixel value of (255,255,255). Therefore, cv2.FILLED parameter value is used to fill the entire area of the contour with a value of (255,255,255) and a binary mask is returned as our final segmentation mask.

Finally, with the help of a bitwise_AND() function, this mask is merged with the pre-processed image. The process of deriving the final mask and the segmented ROI is illustrated in Fig 15.

After implementing the algorithms associated with the segmentation process, it can be observed that the ROIs are successfully extracted as shown in Fig 16.

Table 5 represents all the selected parameter values for the algorithms associated with the segmentation process which are demonstrated above.

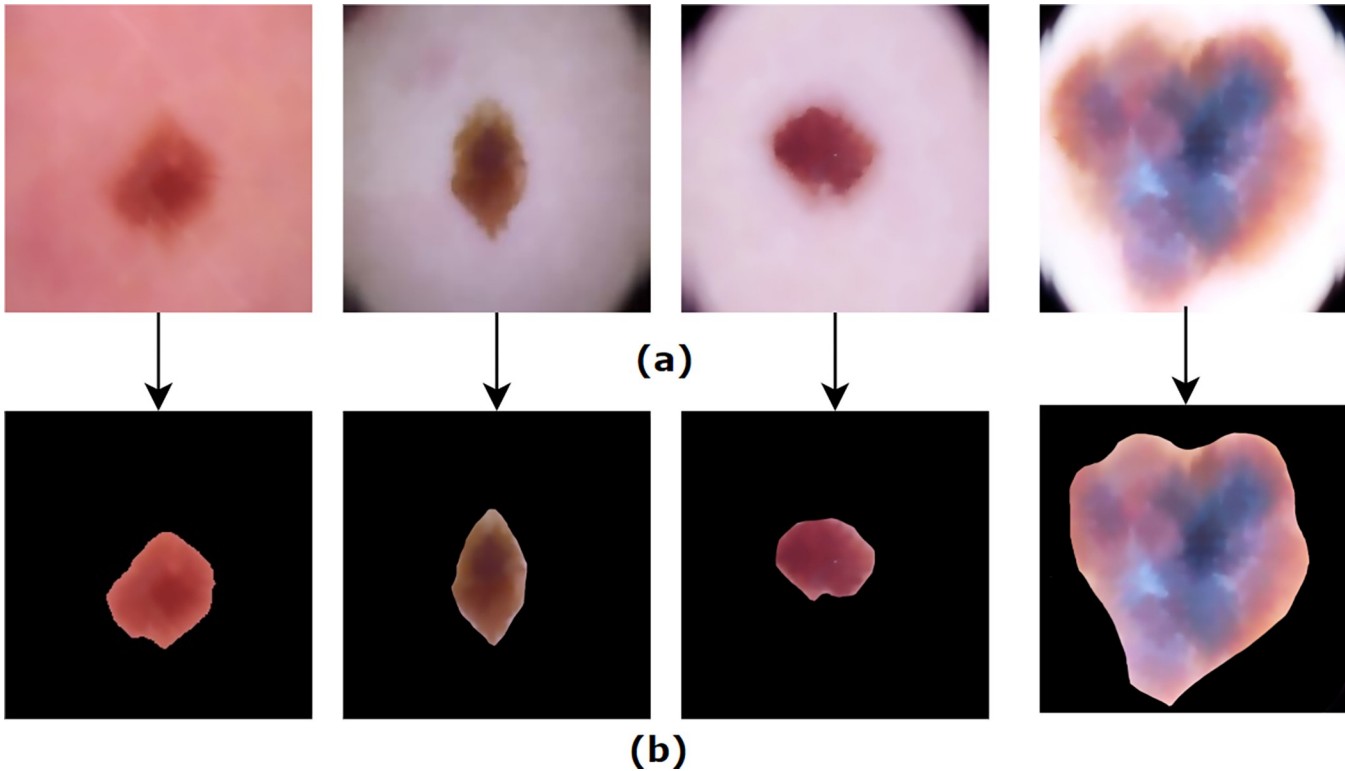

**Fig 16.** (a) Pre-processed images of skin lesion (b) Resultant images after ROI extraction.

## 6.3. Down-scaling

A digital image consists of two-dimensional arrays, having a size (number of pixels) of M x N where the number of rows is denoted by M and the number of columns is denoted by N. The location and value of each pixel are determined by coordinates (x, y) and the intensity value f respectively. Therefore, the representation of a digital image can be described as f(x, y). In this regard, down-scaling is the decrease of spatial resolution, retaining an equivalent two-dimensional (2D) image. This is usually done to reduce the storage requirements of images [33]. Reducing the number of pixels of an image can also contribute to decreasing the training time of a CNN model. As stated, in our study, the term down-scaling means, reducing total space taken by an image without decreasing its actual size. The original resolution of our image was 224x224 and after down-scaling, this will remain the same. But the size of the images before down-scaling was on average around 45 kb. After down-scaling, this size was reduced to

**Table 5. Selected parameter value for segmentation process.**

| Algorithm/process | method | Necessary parameter |
|---|---|---|
| Otsu's thresholding | cv2.threshold() | Threshold value = 120, Maximum value = 255, Thresholding technique = cv2.THRESH_BINARY+cv2.THRESH_OTSU |
| Morphological Dilation | cv2.dilate() | kernel size = (5, 5) |
| Largest blob | cv2.findContours() | Contour retrieval mode = cv2. RETR_EXTERNAL, contour approximation method = cv2.CHAIN_APPROX_SIMPLE |
| | max() | key = cv2.contourArea |
| Hole-filling | cv2.drawContours() | Index = 0, border color = (255, 255, 255), type = cv2.FILLED |

around 6 kb while preserving important details. We have used two filters namely bilateral filter and box blur simultaneously for the purpose. At first, a bilateral filter is applied in order to remove noise, smooth and down-scale pixels. It reduces the image size from 45 kb to 14 kb. Then we apply a box blur filter which reduces the image size from 14 kb to 6 kb. Therefore, the reason for applying two filters is mainly, bilateral filter smooth image by removing noises also reduces size. To reduce size more, box blur is applied. Moreover, after applying box blur, our accuracy did not drop (section 9.5.1). Time complexity, space complexity and result analysis of this approach are demonstrated in section9.5.

**6.3.1. Bilateral filter.** Noise removal techniques can aid to identify details and characteristics of images that are not obvious. This is an important task in biomedical image analysis as the medical images require reliable techniques to get accurate results [34]. The performance of neural networks heavily depends on edge information and noise filtering techniques [35]. The most common de-noising filters described in previous research are mean filter, median filter, Gaussian blur, etc. These filters often cause the loss of important information as they blur out the pixel intensity. To resolve this problem, a bilateral filter, defined as a weighted average of nearby pixels, is introduced. Bilateral filters take into account the difference in the value of neighboring pixels to retain the details while smoothing out the image. The equation of bilateral filter [36,37] is as follows:

Suppose that $\|p_j - p_j\| \in [0, \infty]$ is a spatial distance that can be defined for all index pairs (1: $j$), $11 \leq i, j \leq N$. Then bilateral filter weights [] are stated in Eq 12.

$$w_{ij} = \exp\left(-\frac{\|p_i - p_j\|^2}{2\sigma_d^2}\right) exp\left(-\frac{|g_i - g_j|^2}{2\sigma_r^2}\right) \tag{12}$$

Here $\sigma_d$ and $\sigma_r$ are constant filter parameters, and $|g_i - g_j|$ is the preferable distance to the guidance signal (g) components. The bilateral filter's arithmetical complexity of a signal application to images situated on grids that are rectangular can be reduced to $0(N)$.

Optimal outcome of this filter highly depends on the suitable parameter values. The parameter values can be determined based on the highest PSNR achieved [38]. Therefore, in this study, by tweaking the parameter values and analyzing PSNR, the optimal parameter value is chosen. The algorithm is applied using cv2.bilateralFilter() method, which requires four parameters, input image, diameter of each neighborhood pixel, $\Sigma$ value in color space and $\Sigma$ value in coordinate space. Diameter denotes the diameters of pixel neighborhood or simply filter size. A large filter (d > 5) are used to filter heavy noise but tends to perform slowly, hence we have used d = 5 for our application. A larger sigma value (> 150) of this parameter results in, more pixels within the neighborhood will be mixed together. It causes a strong effect making larger areas under similar pixel intensity. On the contrary, if the value is small (<10), the filter will not have a noticeable effect. Therefore, we have used a value of 75 by experimenting with other values, to achieve the best possible outcome. Alike sigma color, a larger sigma space value (> 150) results in, more pixels within the neighborhood will be mixed together. If the value is smaller (<10), the filter will not have a noticeable effect. Usually, values of Sigma Color and Sigma Space are kept equal. Therefore, a value of 75 is passed as a sigma space value.

The process of applying the bilateral filter is illustrated in Fig 17 where the output images are smoothed without blurring of edges and where the output of gamma correction is used as input of the bilateral filter.

**6.3.2. Box blur.** Down-scaling is carried out using different interpolation algorithms such as bi-linear, bi-cubic and nearest neighbor. These traditional downscaling algorithms often result in losing data and causing irregular outcomes. For this research, a different approach namely box blur algorithm is introduced. The main objective of introducing box blur in our

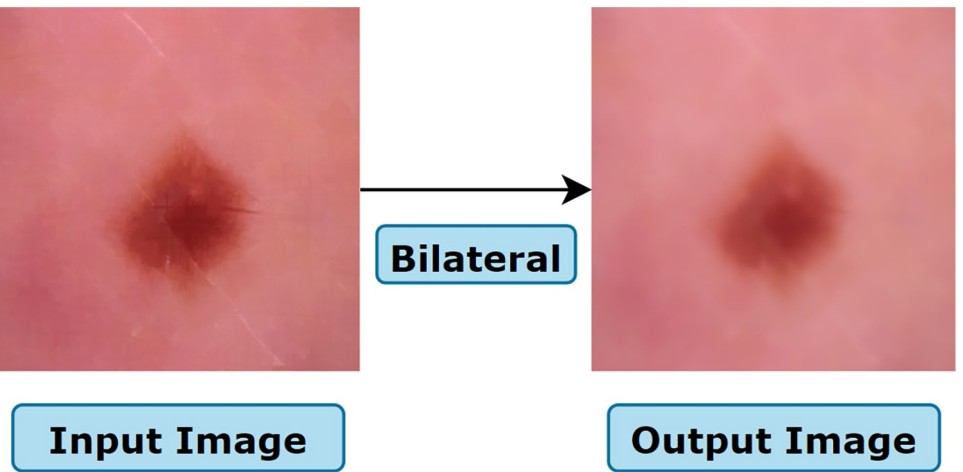

**Fig 17. Resultant image after applying the bilateral filter.**

study is to reduce computational time, storage space and processing power while not sacrificing overall performance. The box filter devised by Crow [39], also known as "moving average" is a simple filter that is recognized for its low computational complexity, optimality and feasible implementation compared to other filters for down-scaling. The importance of box filters in image processing was demonstrated by Viola and Jones with their landmark face detection algorithm [40]. Geavlete et al., [41] applied box filters to accelerate computation for the detection of the Speeded-Up Robust Features (SURF).

Box blur filter is the fastest filter algorithm, using an N x N kernel full of ones. The amount of blurring of the pixels of an image and the computational complexity are determined by the value of N. A weighted average is computed by multiplying the individual pixel with the corresponding kernel matrix. After multiplication, the average of the pixels is calculated. The algorithm for this method is described in Algorithm 5.

```
Algorithm 5. Box Blur.
BEGIN
    1. Let, input image = p(x,y)
    2. Define the kernel size to apply on p(x,y), k = N x N
    3. Total, t = 0
    4. Size of window for the target pixel, w = k // N x N
    5. FOR pixel in w:
        Window pixel, w[i] = w[i] x k[i] //retrieve the values by
multiplying
        t + = w[i] //add all the numbers
    5. Target pixel, Tp = t / k // divide by the area of the kernel
    6. Repeat above for all the remaining windows in x direction
    ENDFOR
END
```

For applying box blur, ImageFilter.BoxBlur() is used. This function requires only one parameter, radius. Radius value 0 does not have any effect on an image. Radius value 1 makes a 1 X 1 window size which has a little effect. Therefore, we used a value of 2 as higher value ($> = 3$) was not providing the expected outcome. Hence, the window size will be $2r \times 2r$, where, r denotes the window radius. The formula for the targeted pixel value using blur box filters

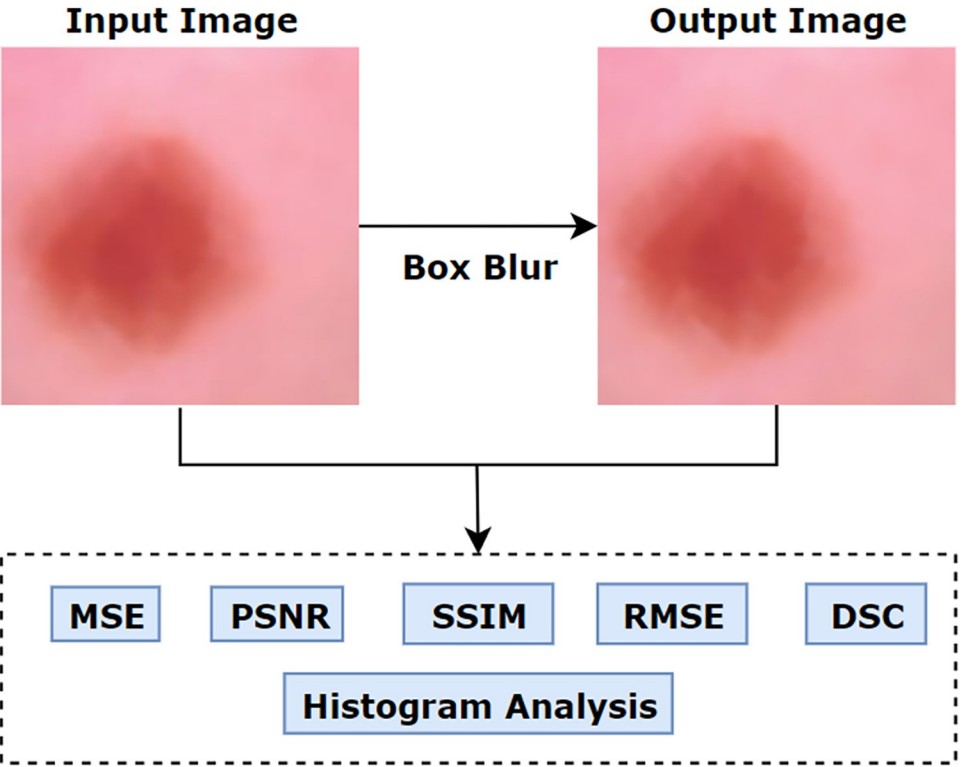

**Fig 18. Resultant image after applying Box blur Table 6 represents all the selected parameter values for the algorithms associated with down-scaling process.**

[42] is stated in Eq 13.

$$P_{i,j} = \sum_{k=0}^{m} \sum_{l=0}^{m} w_{k,l} F_{i+k,j+l} \tag{13}$$

Here, $F_{i,j}$ denotes the pixel values in the image for i, j = 1, . . . . . ., n. Using the box filter, the distributed weights for all the four pixels from which the average of the four pixels is calculated are equivalent.

$$P = \frac{1}{(2r)^2} F_{i,j} + F_{i,j+1} + F_{i+1,j} + F_{i+1,j+1} \tag{14}$$

The resultant image after applying box blur is shown in Fig 18.

Finally, values of MSE, PSNR, SSIM, RMSE, Histogram Analysis and DSC are evaluated for the preprocessed image and the down-scaled image.

**MSE.** MSE is the calculation of cumulative squared error between the pixels of the original and processed image. The range of MSE lies between 0 and 1 where a value close to 0 indicates

**Table 6. Selected parameter for down-scaling process.**

| Algorithm/process | method | Necessary parameter |
|---|---|---|
| Bilateral Filter | cv2.bilateralFilter() | Diameter = 5, Sigma Color = 75, Sigma Space = 75 |
| Box Blur | ImageFilter.BoxBlur() | Radius = 2 |

higher image quality. Formula for MSE is stated in Eq 15.

$$MSE = \frac{1}{mn}\sum_{i=0}^{m-1}\sum_{j=0}^{n-1}\left(G(i,j) - P(i,j)\right)^2 \tag{15}$$

where, G represents the ground truth image (i.e., input image) and P represents the processed image, m and n denotes the pixels of G and P while i, j denotes the rows of m, n pixels.

**PSNR.** PSNR is the ratio between the maximum possible signal power and the power of the distorting noise affecting the quality of a processed image.

$$PSNR = 20 \log 10((MAX)\sqrt{MSE}) \tag{16}$$

where, MAX is the maximum value of the pixels of the image (i.e., 255). The higher the PSNR value, the less the distortion [43]. For an 8-bit image, a good PSNR value typically lies between 30 to 50 decibels.

**SSIM.** In the SSIM method, image degradation is measured as the transformation of perception in structural information [44]. The outcome is in the range of −1 to 1, where 1 signifies 'perfect structural similarity' and 0 signifies 'no similarity'.

$$\text{SSIM}(x, y) = \frac{(2\mu_x\mu_y + c_1)(2\sigma_{xy} + c_2)}{(\mu_x^2 + \mu_y^2 + c_1)(\sigma_x^2 + \sigma_y^2 + c_2)} \tag{17}$$

where, $\mu_x$ and $\mu_y$ are the means of two images $(x,y)$, derived by applying the gaussian window. $\sigma_x^2$ and $\sigma_y^2$ denote the variance, $\sigma_{xy}$ is the covariance of the images and $c_1$ and $c_2$ are the two variables used to alleviate the division. $c_1 = (0.01 \times 255)2$ and $c2 = (0.03 \times 255)2$ where 0.01 and 0.03 are employed as default values.

**RMSE.** RMSE estimates the variance between the predicted value and actual value to assess the magnitude of the error. To compute RMSE, errors are first squared and then averaged. Lowe RMSE especially values close to 0 means good quality.

$$RMSE = \left[\sum_{j=1}^{N}(d_{f_i} - d_d)^2/N\right]^{\frac{1}{2}} \tag{18}$$

Here, $(d_{f_i} - d_d)^2$ is the square of differences of predicted value $(d_{f_i})$ and actual value $(d_d)$ and $N$ denotes the size of the dataset.

**DSC.** The DSC is computed as a statistical validation metric to estimate the similarity of images before and after applying box blur filter. The range of DSC values is from 0 to 1, where a value close to 1 point to a higher resemblance of the images.

$$DSC = \left(\frac{2 * \text{Area of Overlap}}{\text{Total Number of Pixels in both Images}}\right) \tag{19}$$

Table 7 represents MSE, PSNR, SSIM, RMSE and DSC values of 10 randomly chosen images.

Table 7 states that PSNR values of 10 blur box processed images are in the range of 40–43 dB, indicating that the images are of good quality. SSIM values are in the range of 0.90 to 0.96, which demonstrates that the images are structurally similar even after the application of downscaling using the blur box algorithm [45].

**Histogram analysis.** The histogram of an image is a graph that signifies the intensity amounts for every pixel. In this graph, the x-axis represents the intensity value in a range of 0–255 and the y-axis represents the number of pixels. Comparing the histograms of two

**Table 7. MSE, PSNR, SSIM, RMSE and DSC of 10 images.**

| Image | MSE | PSNR | SSIM | RMSE | DSC |
|---|---|---|---|---|---|
| Image_1 | 3.296 | 42.949 | 0.953 | 0.08 | 0.983 |
| Image_2 | 3.876 | 42.231 | 0.939 | 0.09 | 0.971 |
| Image_3 | 4.310 | 41.432 | 0.912 | 0.10 | 0.965 |
| Image_4 | 3.564 | 42.120 | 0.922 | 0.09 | 0.974 |
| Image_5 | 4.450 | 41.032 | 0.910 | 0.11 | 0.960 |
| Image_6 | 3.345 | 42.865 | 0.949 | 0.08 | 0.973 |
| Image_7 | 3.899 | 42.110 | 0.922 | 0.08 | 0.970 |
| Image_8 | 4.810 | 40.432 | 0.906 | 0.12 | 0.958 |
| Image_9 | 4.834 | 40.128 | 0.900 | 0.12 | 0.956 |
| Image_10 | 4.410 | 41.011 | 0.913 | 0.11 | 0.968 |

images, changes can be evaluated. In our study, a histogram of an original image and an image after applying box blur filter are generated (Fig 19).

## 7. Augmentation and training experiments

Data augmentation techniques are applied to increase the number of images. The augmented dataset is split into training, validation and testing. After splitting, the training dataset is used to train our proposed SCNN_12 model by performing an ablation study to determine the optimal parameters, hyper-parameters and their values based on the highest classification accuracy. Fig 20 illustrates the complete process.

### 7.1. Augmentation

CNNs have been demonstrating good performance in computer vision tasks, including automated skin lesion detection and classification over the decades. To build deep learning models, the validation error must diminish with the training error. However, these models often tend to overfit and require ample size datasets, unfortunately the availability of annotated skin lesion images is somewhat inadequate and annotation is arduous and time-consuming. However, data augmentation can aid to expand datasets by transforming and over-sampling images while retaining original pixel details and target labels. A fundamental requirement of this method is that the sematic meaning remains unchanged in the generated images. This technique has been adopted in skin lesion classification tasks to reduce overfitting issues [46]. In

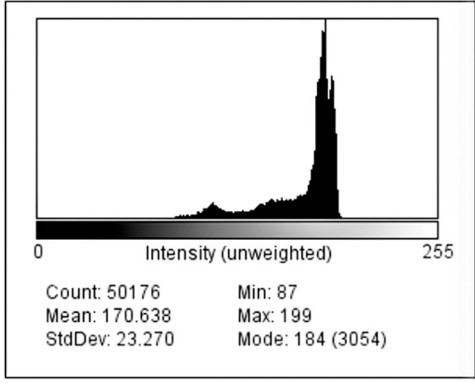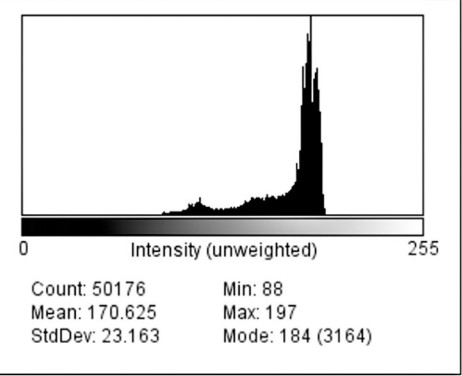

**Fig 19. Histograms of original image vs box blur filter.**

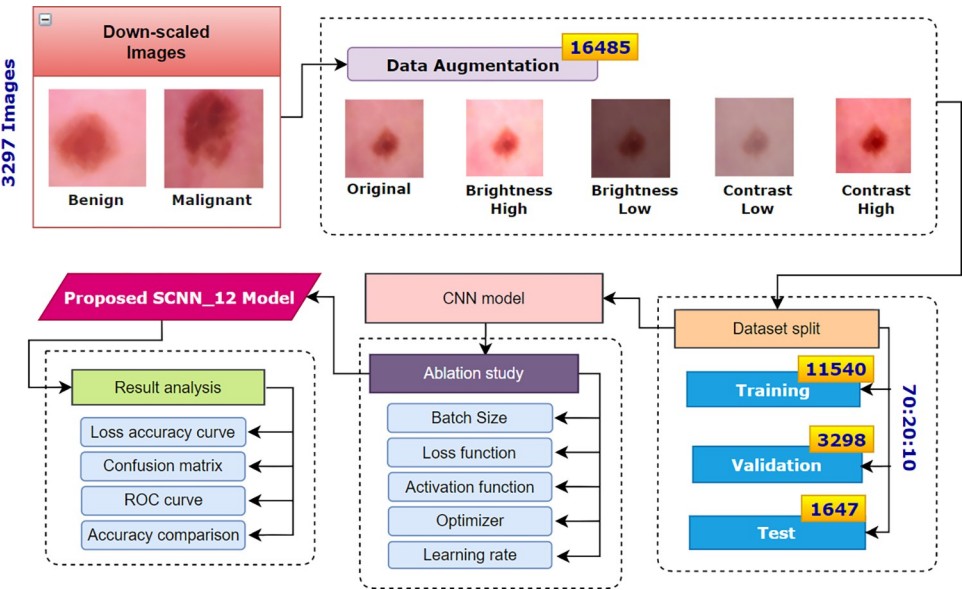

**Fig 20. Augmentation and classification process.**

our study, use of data augmentation is employed to overcome the problem of data inadequacy. Previous research has demonstrated the effectiveness of data augmentations by simple transformations such as flipping, rotation, color space augmentations, zooming and random cropping. However, changing the color distribution of images by decreasing or increasing the pixel intensity can be an effective alternative way to augment the data [47,48]. In some cases, geometrical augmentation methods actually decrease the performance, as the structure of the medical image is mostly complex [47]. These transformations are very useful on straightforward images. A few shortcomings of geometric conversions are, required additional memory resources, higher computation costs and training time [47]. Moreover, in several research domains such as medical data analysis, the biases distancing the training data from the testing data are more complex than positional and translational variances. Therefore, the scope of where and when geometric transformations can be applied is relatively limited. In medical imaging, the location of a tumor is very important in detecting abnormalities. Geometrical augmentation for instance flipping or rotating can change the exact location of the tumor or affected area. Therefore, this may affect on overall accuracy. On the contrary, photometric transformation augments images just by altering the intensity value, which can be an effective approach in the medical domain. This technique balances image lighting and color while keeping the geometry unaffected. Therefore, instead of using typical transformation techniques such as flipping, rotation or cropping; brightness [20] and contrast alteration of images is carried out for data augmentation.

Brightness is a relative term defined as the overall lightness or darkness of an image. On the other hand, contrast is the difference of brightness between the object and background presented in an image. The mathematical formula can be expressed as:

$$g(x) = s(x) + \beta \tag{20}$$

here, s(x) = source image pixels and g(x) = output image pixels after altering brightness. Increasing or reducing the value of parameter $\beta$ will add or deduct a constant value to every pixel. A positive $\beta$ value causes brightened image, whereas a negative value results in darkening

the image. To enhance the contrast level, the brightness difference needs to be increased by multiple. The mathematical formula can be expressed as:

$$g(x) = \alpha \times s(x) \tag{21}$$

In this formula, s(x) = source image pixels and g(x) = output image pixels after altering contrast. The parameter α >1 means enhancing contrast and α <1 means generating an image of less contrast. Similarly, β >1 means enhancing brightness and β<1 means decreasing brightness. For this study, we experimented with various beta (β) and alpha (α) values and choose alpha values 1.3 and 0.7 for respectively bright and darkened images and beta values 1.3 and 0.7 respectively for high contrast and low contrast image. Based on the above formulas, the dataset is augmented byapplying four approaches, increasing brightness, decreasing brightness, increasing contrast and decreasing contrast. As shown in Fig 20, after data pre-processing and applying down-scaling, 3297 down-scaled images were obtained. Four augmentation techniques have been employed on both datasets before and after applying down-scaling methods.

In addition, to assess the effectiveness of our photometric augmentation approach, some other geometric augmentation possibilities are explored. In this regard, data augmentation is carried out by horizontal flip, vertical flip, rotation by 90˚ and 45˚ angle. Both photometric and geometric augmented datasets are trained in order to evaluate the comparison between them in terms of accuracy. The experimental result of this comparison is showed in section 9.6 Table 15.

## 7.2. Data split

The last step before conducting the training process of the model involves splitting the dataset. In the training phase, the model is trained with a dataset and validated with a separate dataset. Afterward, the learning weights are saved and tested on another test dataset of unknown images. Hence, a dataset must be split in training, validation and testing datasets. Generally, three common splitting ratios for training-testing data (90:10, 80:20, and 70:30) are used to assess the model's learning [49]. We split both of the augmented datasets into three sets following the ratio of 70:20:10 for training, validation and testing respectively. After splitting the augmented dataset of total 16,485 dermoscopy images, we get 11,540 images in the training set, 3298 images in the validation set and 1647 images in the testing set, all including both benign and malignant cases which is shown in Table 8.

## 7.3. Ablation study

At present, the method "ablation study" is widely used in the context of neural networks with an objective to attain in-depth perception of the network's performance by examining the consequence of altering or removing some elements [42]. When different components, parameters and hyper-parameters of the architecture is removed or altered, the model tends to yield

Table 8. Number of images in classes after spitting.

| Dataset | Class | Number of images |
|---|---|---|
| | Benign | 6300 |
| Training set | Malignant | 5240 |
| | Benign | 1800 |
| Validation set | Malignant | 1498 |
| | Benign | 900 |
| Testing set | Malignant | 747 |

decreased, identical or increased performance. Based on the study, any potential flaws of the model can be detected and resolved by upgrading the architecture. In our research, the parameters and hyper-parameters of the proposed model are determined by performing an ablation study based on the optimal performance in terms of accuracy. In building a most effective CNN model for a particular dataset, it is important to experiment with different parameters and changing components of the model. Keeping this in mind we employed ablation study by tweaking the parameters and changing components of CNN such as batch size, loss function, activation function, optimizer and learning rate. In this regard, four study cases are employed by exploring different CNN components. The results of ablation study are explained in Section 9.2.

**7.3.1. A detailed explanation of ablation study that have been used for the experiment.** Several CNN components such as learning rate and batch size, [50] optimizer, loss function can be varied to improve the performance of a CNN model. In ablation study 1, the model is trained using three different batch sizes (16,32,64) [20] and the optimal batch size is chosen based on highest accuracy. In ablation study 2, two output layer activation functions namely softmax and sigmoid are employed to determine for which, highest accuracy is gained. Four widely-used loss functions namely, Binary Cross Entropy, Mean Squared Error [51], Categorial Cross Entropy and Kullback-Leibler Divergence [51] are experimented in case study four as changing loss function impacts on overall performance. Finally, the network is trained six times on the down-scaled augmented dataset with three different optimizers namely Adam, Nadam and Adamax using two different learning rates of 0.001 and 0.001. In this process, the model is trained on the training dataset and validated on the validation data set with each of the learning rates and optimizers independently. Afterward, each model is tested with the test set to find the best model in terms of accuracy.For the pre-processed augmented dataset (before down-scaling) the model is trained only with the best optimizer and learning rate to compare the time complexity and overall performance of the model for the datasets before and after applying down-scaled method.

## 8. Model

Traditional CAD tools, which require extracting features manually, are found to have major drawbacks [52] as custom-made features are mostly domain specific with an arduous process of feature design. An alternative and possibly superior method is to learn the significant features from images directly and automatically through a CNN model without any human interpretation. CNNs are designed to utilize spatial and configurational layers, interspersed with pooling layers, followed by fully connected layers which together capture the spatial and temporal features of an image, through relevant filters. This is very effective in reducing the number of parameters without reducing the quality of the model. CNNs can be trained to understand a sophisticated image dataset and increasingly applied in medical image classification, detection and segmentation tasks because of its high accuracy and the reusability of weights. We, therefore, propose a narrow-depth CNN model for classifying skin lesions into two classes: benign and malignant.

CNN is a complex feed-forward variant of neural networks. The architecture is composed of multiple layers of artificial neurons. These work as a placeholder of a mathematical function where a weighted sum of a given input and a predicted output is provided. Thus, the behaviors of all neurons are defined by their weights. The neurons learn in each layer learn the weights by a back propagation algorithm.

CNNs are made up of three key layers namely (a) convolutional layers which extract various features from the input images and provide an output called the feature map. (b) pooling layers

that decrease the size of the convoluted feature map to reduce the computational costs and finally (c) fully connected layers (FC) which consist of weights and biases and neurons. This has been implemented in computer vision tasks, originally to detect low level features (such as structure, edges and curves) and later to discover more abstract characteristics. The input x of a CNN model has three dimensions: h×w×d, where h is the height, w is the width and d is the depth or number of channels. In this research, the height and weight are equal or h = w. The convolution layer computes a dot product between the input and weights to generate feature maps by introducing nonlinearity with an activation function:

$$h^k = f(W^k * x + b^k) \tag{22}$$

Here, $h^k$ are the feature maps to be generated, $b^k$ is the bias, $W^k$ are the weights and $x$ is the input image [53].

Pooling layers reduce the number of parameters which aids to diminish the computational cost and accelerate the training process, adjusting for over-fitting by functioning individually on every depth wedge of an input image. The FC layer connects the neurons of two different layers and is normally placed before the output layer of CNN architecture. It is convenient to put a non-linear activation function on each convolutional layer where CNNs with rectified linear units perform better with lower computational load than CNNs with equivalent tanh units [54]. The mathematical representation can be defined as:

$$f(x)_{ReLU} = max(0, x) \tag{23}$$

This function can be expressed as [26]:

$$U(y) = max(0, y) = \begin{cases} y & \text{if } y \geq 0 \\ 0 & \text{if } y < 0 \end{cases} \tag{24}$$

The dropout technique is adopted to discard neurons randomly in the FC layer causing the model to learn diverse independent features. The hidden layer and all these layers are stacked together to build the CNN architecture.

A very important parameter of CNN network is the Activation Function which controls if a neuron should be activated or not. This function may vary from one hidden layer to another and is responsible for introducing non-linearity into the output of a neuron. The Softmax activation function [55] is often used in solving both binary and multi-class classification problems. However, the main objective of CNN is minimizing the loss function using Eq 25 [56]:

$$\begin{aligned} L &= \frac{1}{2} \sum_{i=1}^{n} (\|Y_i - S_i\|^2) \\ &= \frac{1}{2} \sum_{i=1}^{n} (\|Y_i - W_i X_i\|^2) \end{aligned} \tag{25}$$

here, n represents the total number of training observations and $\sum_{i=1}^{n}$ indicates the sum over all samples. Alongside, $Y_i$ denotes the true value for all samples and all predicted outcomes are symbolized by $S_i$ where $X_i$ is the frequency of training vectors with corresponding weights ($W_i$). The difference between these actual and predicted values is calculated over all instances to complete the further investigation.

The output of the softmax function in our case is a probability p ∈ {0, 1} which is applied as the substitution of the square error loss function. The mathematical expression of the

predicted output class probability is [53] is:

$$p_i = \frac{e^{a_i}}{\sum_{k=1}^{N} e_k^a} \qquad (26)$$

here, $e^{a_i}$ denotes the non-normalized outputs from the previous layer and the inputs of the softmax function. N refers to the number of neurons in the output layer. Softmax normalizes all the preceding values to the range of 0 to 1 where the addition of these values always equals 1. Thus, if the probability of one particular class changes, it affects the other values as well so that the summation of probabilities remains 1.

CNNs have a complex network structure comprising a number of layers, typically requiring massive computation and memory resources, and a large time complexity. During training, as the data distribution of each input in the central layers tends to be distinct, the parameters are updated, leading to a high computation time [57]. A shallow architecture is more applicable for small datasets than deep CNN models as complex architectures tend to cause overfitting issues with a small number of images [58]. To avoid these issues, we construct a shallow CNN network with few layers small convolution kernel sizes to achieve satisfactory performance along with a relatively low computational cost [59]. For a shallow architecture, assuming that size of the input feature map is $H \times W \times N$, the size of the output feature map is $H \times W \times M$ and the convolution filter size is $K \times K$, the number of parameters of a standard convolution layer will be $K2 \times N \times M$, and the number of parameters of a depth wise distinguishable convolution layer can be calculated as $K2 \times N + 12 \times N \times M$. The result shows that the depth wise discrete convolution is $1/M + 1/K2$ for the standard convolution parameters only.

## 8.1. SCNN_12 architecture

Our proposed model is composed of 12 weighted layers: 4 convolutional layers followed by 4 max-pooling layers, a flatten layer, two dense layers and a softmax layer as shown in Fig 21.

Each block in the architecture constitutes one $2 \times 2$ kernel sized convolutional layer followed by one max pooling layer and a Rectified linear unit (ReLU) activation function for features extraction. The input shape is defined as $224 \times 224 \times 3$ where height x width = 224 x 224 and 3 denotes the number of color channels (RGB format). The convolution layers in each

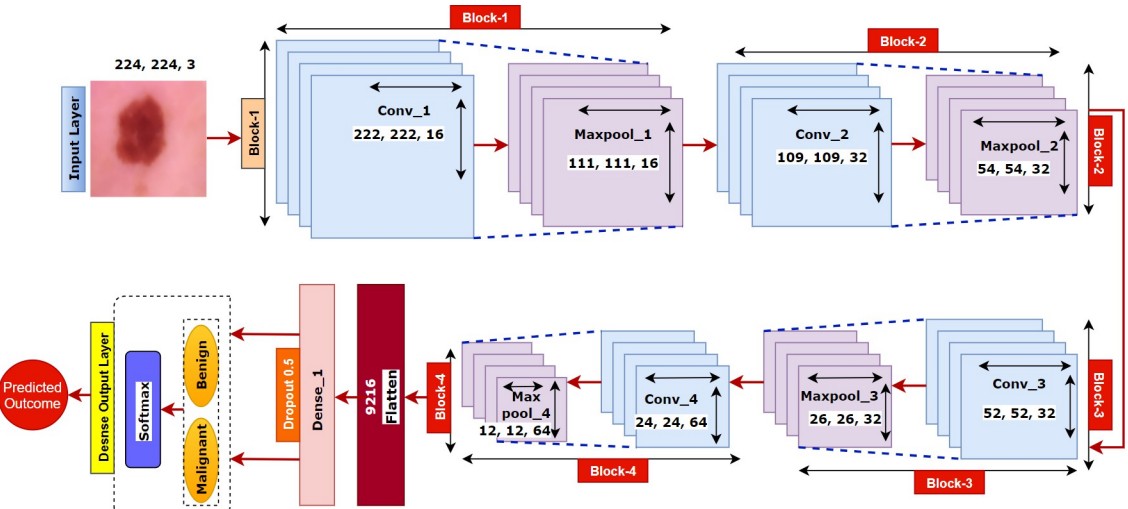

**Fig 21. Architecture of proposed SCNN_12 model.**

block extract complex data features and the maxpool layer reduces the extracted features. Max pool layer reduces the feature map to lessen the computational complexity while preserving the relevant features of the input image. First, the input layer is fed to block-1 containing the first convolution layer of 16 filters with kernel size $3 \times 3$. This extracts the features from the input image. This is then followed by a $2 \times 2$ max pool layer which shrinks the extracted feature map from the first convolution layer into half its size ($222 \times 222$ is reduced to $111 \times 111$). Block-2 and block-3, consist of a $3 \times 3$ kernel sized convolution layer of 32 filters, followed by a maxpool layer of $2 \times 2$. These blocks also extract features from the resultant feature map of block-1 and reduces the size of the image half to its size. Block-3 results in a much smaller feature map with dimension $52 \times 52$. Block-4 consists of the fourth convolutional layer of 64 filters with an equivalent size of kernel $3 \times 3$ and a $2 \times 2$ max pooling filter. Block-4 produces a feature map of dimensions $12 \times 12$. Block-4 is followed by a flatten layer which converts the pooled feature map into a 1D vector. The flatten layer is followed by a FC layer comprising 1024 neurons equipped with the ReLU activation function. A dropout layer with a value of 0.5 is placed after the FC layer to deal with over-fitting issues and to accelerate the training. As the model is designed to classify the images into two classes, the classification layer or output layer consists of a FC layer with two neurons which takes the 1D resulting tensor as input and gives a binary output where 0 and 1 respectively indicate benign and malignant class.

## 8.2. Training procedure

After conducting ablation study, the proposed model is trained with a batch size of 32 and the maximum number of epochs is set to 200. Weights based on the minimum loss value are saved using the 'callback' function of Keras during the training process. 'Categorical cross-entropy', specified as 'categorical_crossentropy' in Keras, is used as the loss function when compiling the model. The cross-entropy loss function is typically applied to the feature discriminate network. The relevant equation is as follows [23]:

$$L = \begin{cases} -y\ln p & , y = 1 \\ -(1 - y)\ln (1 - p), & y = 0 \end{cases} \qquad (27)$$

$$FL = \begin{cases} -\alpha(1 - p)^{\gamma}y\ln p & , y = 1 \\ -(1 - \alpha)p^{\gamma}(1 - y)\ln (1 - p), & y = 0 \end{cases} \qquad (28)$$

$$L_{\text{total}} = \begin{cases} AL + BL + CL, & \text{use } L \\ AFL + BFL + CL, & \text{use } FL \end{cases} \qquad (29)$$

Where $L$ denotes the cross-entropy loss function, $y$ represents the actual category label, and $p$ refers to the predicted category label probability value. In Eq 28, $FL$ is the focal loss function, $\alpha$ is the factor applied to balance the impact of negative and positive classes to the loss function value and another factor $\gamma$ is employed to assess the effect of rigid and easy samples to the loss function value. In Eq 29, $L_{\text{total}}$ is a measure of the entire loss of the training model, $A, B$ signifies the loss function weight of the two classification models in the total loss function and C denotes the loss function weight of the feature categorize network in the overall loss function. The values of A, B, and C are initialized to 1, 1, and 0.5 respectively. Tensor flow, a well-known framework that provides Application Programming Interfaces (APIs), is used for building and evaluating the model. The model is trained on Jupyter notebook with Anaconda's latest python version 3.8. Since the training of neural network is computationally intensive, especially with a

large dataset, it is crucial to utilize Graphical Processing Units (GPU). Three computers are used for this research equipped with Intel Core i5-8400 Processor, NVidia GeForce GTX 1660 GPU, 16 GB of Memory, and 256 GB DDR4 SSD for storage.

## 9. Result and discussion

To evaluate the proposed SCNN_12 classification model several metrics are used namely accuracy, precision, recall, fi-score, and specificity. In order to assess AUC [60] value, ROC curves are generated for all configurations. The model evaluation error metrices: mean absolute error (MAE) and root mean squared error (RMSE) are also evaluated. Confusion matrix generates the values of true positive($TP$), true negative($TN$), false positive($FP$), false negative($FN$). The evaluation matrices are calculated with the values obtained from the confusion matrix. In this section, all the ablation study cases with results are discussed along with statistical analysis.A comparison of the time and space complexity is presented to evaluate the efficiency of our introduced down-scaling method. Space complexity is found out by comparing the storage taken by images before and after applying dawn-scaling method. Finally, results of k-fold cross-validation, splitting dataset with different ratios and testing the model with noise enduced dataset are shown.

### 9.1. Evaluation metrices

According to [61], a classifier's capability to accurately predict the classes is considered the main measure in evaluating a binary classification model. This section describes the evaluation metrices used to evaluate the quality of the CNN classification model.

Accuracy (*Acc*) is a measure to evaluate the model's capability to predict the correct class in the test set. It is one of the simplest evaluation metrics. It is calculated as the sum of TP and TN divided by the sum of TP, TN, FP and FN.

$$ACC = \frac{TP + TN}{(TP + TN + FP + FN)} \tag{30}$$

Recall, the metric that evaluates the ability of a model to predict the true positive rate. It is the ratio of true positives to all positive predictions in the test set. It is calculated by dividing TP by the sum of TP and FN

$$Recall = \frac{TP}{(TP + FN)} \tag{31}$$

Specificity indicates the model's capability to predict the true negative rate. It is found by dividing TN by the sum of TN and FP

$$Specificity = \frac{TN}{(TN + FP)} \tag{32}$$

The precision metric quantifies the proportion of correctly predicted positive cases made in the test set. It is the ratio of TP to all the positive predictions of the classifier (*TP+FP*).

$$Precision = \frac{TP}{(TP + FP)} \tag{33}$$

F1-score (F1) produces a single score that balances the concerns of both precision and recall in just one number. In other words, it gives an overall performance measure of the

classification model. It is calculated with the following equation,

$$F_1 = 2 * \frac{Precision * Recall}{(Precision + Recall)} \tag{34}$$

MAE analyzes the average of the error, where error is the absolute difference between the actual values and predicted values, for each sample in a dataset.

$$MAE = \sum_{i=1}^{n} |x_i - x_t| \tag{35}$$

here n denotes the number of total observations in the test dataset $x_i$ is the true values of the test set and $x_t$ is the value predicted by the classifier.

RMSE is the square root of the mean of the square of all errors, providing the average model prediction error.

$$RMSE = \left[ \sum_{j=1}^{N} (d_{f_i} - d_d)^2 / N \right]^{\frac{1}{2}} \tag{36}$$

here N denotes the number of total observations in the test dataset $d_{f_i}$ is the true values of the test set and $d_d$ is the value predicted by the classifier.

ROC curve is sn evaluation metric, which is obtained by plotting the True Positive Rate (*TPR*) or Recall against False Positive Rate (*FPR*) at several threshold values. *FPR* is calculated as:

$$FPR = 1 - Specificity = 1 - \frac{TN}{TN + FP} \tag{37}$$

## 9.2. Performance analysis based on ablation study

The augmented skin cancer dataset is used to train our proposed SCNN_12 classification model based on ablation study. In this context, the results recorded for all the case studies are demonstrated.

**9.2.1 Ablation study 1: Changing batch size.**   In this study, we have experimented with three batch sizes 16, 32 and 64 with optimizer Adam and learning rate 0.001. Table 9 is evident that batch size 32 is giving the highest test accuracy (98.43%) among all the batch sizes while also having a high training and validation accuracy.

**9.2.2 Ablation study 2: Changing output layer activation function**. In the output layer, we experimented with both Softmax activation function and Sigmoid activation function. While both of these perform quite well, it is observed from Table 10 that Softmax activation function gave the highest test accuracy (98.74%).

**9.2.3 Ablation study 3: Changing loss function.**   Experimentation with various loss functions was employed in ablation study. Loss functions: binary cross entropy, Mean squared error, Categorial Cross Entropy and Kullback-Leibler Divergence were chosen for experimentation. Our findings of Table 11 indicate that on the augmented dataset, both binary cross

**Table 9. Performance analysis by changing batch size.**

| Case study | Batch size | Training Accuracy (%) | Validation Accuracy (%) | Test Accuracy (%) | Findings |
|---|---|---|---|---|---|
|  | 16 | 98.73 | 97.54 | 98.32 | Less accuracy |
| 1 | 32 | 98.94 | 97.96 | 98.43 | Highest accuracy |
|  | 64 | 98.67 | 97.65 | 98.17 | Less accuracy |

**Table 10. Performance analysis by changing output layer activation function.**

| Case study | Output layer activation function | Training Accuracy (%) | Validation Accuracy (%) | Test Accuracy(%) | Findings |
|---|---|---|---|---|---|
| 2 | Softmax | 98.73 | 97.54 | 98.74 | Highest accuracy |
| | Sigmoid | 97.16 | 95.22 | 96.75 | Less accuracy |

entropy and Categorial cross entropy yield a similar performance obtaining the identical accuracy (98.74%) of the bunch.

**9.2.4. Ablation study 4: Changing optimizers and learning rate.**    Tables 12 and 13 show the performance of the SCNN_12 model configured for three different optimizers (Adam, Nadam and Adamax) with learning rates 0.001 and 0.0001. The 2×2 confusion matrix was generated for all configured models to calculate the performance metrics for the comparison of these six configured SCNN_12 models.

In Table 12, the optimizer is referred to as 'OP', the learning rate as 'LR', the training accuracy as 'T_acc', the training loss as 'T_Loss', the validation loss as 'V_Loss', the validation accuracy as 'V_Acc', the test loss as 'Te_Loss' and the test accuracy as 'Te_Acc'. It can be observed from Table 12 that optimizer Adam with a learning rate of 0.001 performs best, recording the highest test accuracy of 98.74%. it also gained a validation accuracy of 99.49%. This configuration also has the lowest validation and test loss of 0.02 and 0.05 respectively. In contrast, Adamax with a learning rate of 0.0001 has the lowest validation and test accuracy of 98.57% and 97.42% respectively. The rest of the models have a moderate performance with validation accuracies above 98% and validation losses lower than 0.06. In terms of training accuracy and loss, all the training accuracies were in the range of 98–99% and most of the losses were below 0.04 which indicates that there are no signs of overfitting or underfitting issues.

In Table 13 the optimizer is denoted as 'OP', the learning rate as 'LR', specificity as 'Spe' and precision as 'Pre'. Considering three optimizers, Adam with a learning rate 0.001 yields the most accurate prediction with an AUC of 98.65%, recall of 99.11%, specificity of 98.05%, precision of 98.94% and f1-score of 99.02%. The poorest performance is obtained by Adamax with a learning rate of 0.0001 with an AUC of 97.33%, recall of 97.85%, specificity of 96.83%, the precision of 97.63% and an F1 score of 97.74%. Optimizer Nadam with a learning rate 0.001 performed adequately, achieving scores close to the best configuration (Adam 0.001).

## 9.3 Statistical analysis

Root Mean Squared Error and Mean absolute error matrices were used to evaluate all six configurations of the proposed SCNN_12 model. Comparison between different outcomes based on root mean squared error is represented in Fig 22.

Fig 22 shows the RMSE for all six configurations. For RMSE, the smaller the error value, the better is the performance. It is observed that Adam with a learning rate of 0.001 demonstrates the lowest RMSE whereas Adamax with a learning rate of 0.0001 has the highest RMSE. The remainders of the configurations of SCNN_12 perform moderately recording values in the range of 25.2–31.5.

**Table 11. Performance analysis by changing loss function.**

| Case study | Loss function | Training Accuracy (%) | Validation Accuracy (%) | Test Accuracy (%) | Findings |
|---|---|---|---|---|---|
| | **Binary cross entropy** | **98.73** | **97.54** | **98.74** | **Highest accuracy** |
| 3 | Mean squared Error | 95.64 | 93.28 | 93.58 | Less accuracy |
| | **Categorial Cross Entropy** | **98.73** | **97.54** | **98.74** | **Highest accuracy** |
| | Kullback-Leibler Divergence | 95.64 | 94.28 | 94.87 | Less accuracy |

**Table 12. Training loss(T_loss), training accuracy (T_acc), validation loss (V_loss), validation accuracy (V_acc), test loss (Te_loss), test accuracy (Te_acc) for each of the optimizers corresponding to the learning rates.**

| OP | LR | T_Acc | V_Loss | V_Acc | Te_Loss | Te_Acc | Findings |
|---|---|---|---|---|---|---|---|
| **Adam** | **0.001** | **99.66** | **0.021** | **99.49** | **0.056** | **98.74** | **Highest accuracy** |
| | 0.0001 | 99.35 | 0.035 | 99.20 | 0.061 | 98.14 | Less accuracy |
| Nadam | 0.001 | 99.51 | 0.031 | 99.29 | 0.057 | 98.32 | Less accuracy |
| | 0.0001 | 99.31 | 0.368 | 98.92 | 0.068 | 97.89 | Less accuracy |
| Adamax | 0.001 | 99.06 | 0.041 | 98.73 | 0.073 | 97.65 | Less accuracy |
| | 0.0001 | 98.99 | 0.066 | 98.57 | 0.081 | 97.42 | Less accuracy |

Afterwards, comparison between different outcomes based on mean absolute error is represented in Fig 23.

The lower the MAE value, the better the performance. It can be observed from Fig 23 that Adam with a learning rate 0.001 has the lowest Mean Absolute Error MAE of 0.816 whereas Adamax with a learning rate of 0.0001 has the highest MAE of 1.381. The remainder of the configurations performs moderately, recording values in the range of 1.11–1.31.

After conducting ablation study of four test cases, the configuration of the resulted model is shown in Table 14.

## 9.4. Result analysis of best model

For our best configured SCNN_12 model (Adam 0.001) based on the highest test accuracy, the training and validation accuracy and loss curves were generated as well as the ROC curve and the confusion matrix.

The training accuracy and training loss curve illustrate how efficiently the model is learning from the training dataset whereas the validation accuracy and validation loss curve represent how successfully it is applying its learning to an unknown validation dataset over the training phase. Figs 24 and 25 represent the loss and training curve respectively for our best configured model (Adam 0.001). As can be seen from Fig 25, the training curve is converging smoothly from the very first epoch to the last epoch without major bumps. As shown in Fig 25, the validation curve also does not have any major ups and downs till the endpoint. Validation accuracy rises over epochs along with the training accuracy and the gap between the validation accuracy curve and the training accuracy curve is negligible which indicates no occurrence of overfitting during training of the model. Like the accuracy curve, the loss curve of Fig 24 converges steadily to the final epoch with diminishing loss values. The lower the loss value, the higher the accuracy, and both Figs 24 and 25 verify the efficiency of the proposed model (SCNN_12 with Adam 0.001) in terms of training and validation.

The confusion matrix for the best model is represented in Fig 26. The rows denote the actual label of the test images and the columns denote the predicted label of test images after

**Table 13. AUC, Recall (Rec), Specificity (Spe), Precision (Pre) and F1-score (F1) for each of the optimizers corresponding to the learning rates.**

| Op | LR | AUC | Rec | Spe | Pre | F1 |
|---|---|---|---|---|---|---|
| **Adam** | **0.001** | 98.65 | **99.11** | **98.05** | **98.94** | **99.02** |
| | 0.0001 | 97.93 | 98.55 | 97.60 | 98.35 | 98.45 |
| Nadam | 0.001 | 98.26 | 98.84 | 97.82 | 98.54 | 98.69 |
| | 0.0001 | 97.61 | 98.21 | 97.26 | 98.11 | 98.16 |
| Adamax | 0.001 | 97.39 | 98.01 | 97.11 | 97.87 | 97.94 |
| | 0.0001 | 97.33 | 97.85 | 96.83 | 97.63 | 97.74 |

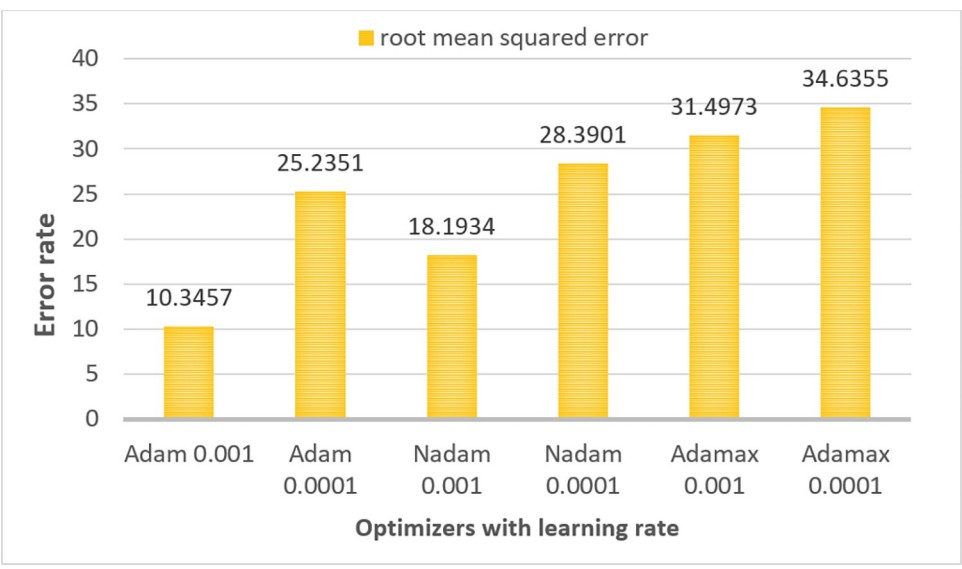

**Fig 22. Root mean squared error of all six configurations of SCNN_12.**

training. As we are dealing with a binary classification problem, two classes can be seen in the matrix: benign (BC) and malignant (MC). Here, the diagonal values indicate successfully predicted TP and TN values. The matrix shows no major bias to any one class predicting the classes quite consistently. By classifying 98.74% of the test images correctly, our proposed SCNN_12 model achieves the best performance.

The Receiver Operator Characteristic (ROC) probability curve is plotted and the AUC value is derived from the ROC curve. The AUC is used as a summary of the ROC curve representing the performance of a model in distinguishing the positive and negative classes. An AUC value close to 1 indicates that the model is capable of detecting almost all the TP and TN values flawlessly. From Fig 27, it can be seen that the ROC curve almost touches the top of the

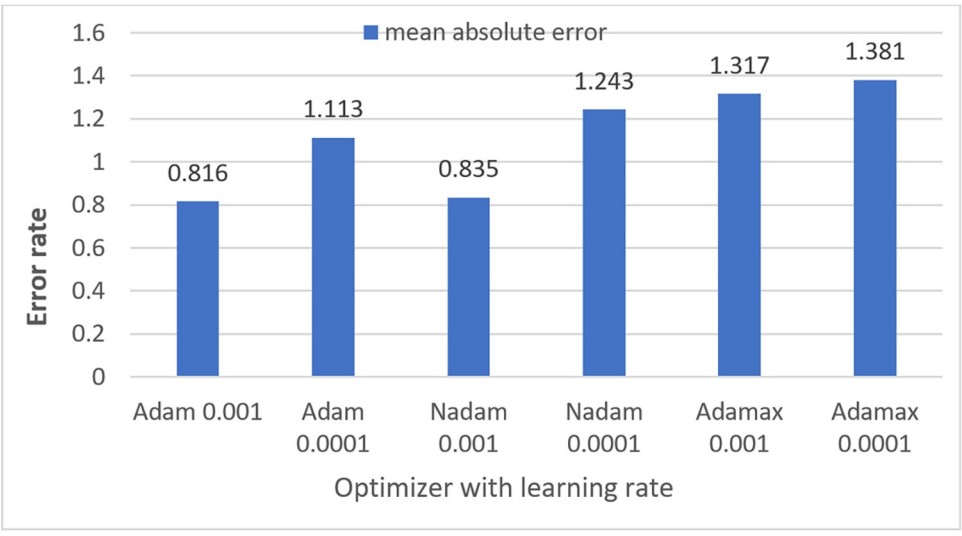

**Fig 23. Mean absolute error of all six configurations of SCNN_12.**

**Table 14. Configuration of optimal model based on hyper-parameters.**

| Configuration | Value |
|---|---|
| Input shape | $224 \times 224$ |
| Epochs | 200 |
| Optimization function | Adam |
| Learning rate | 0.001 |
| Batch size | 32 |
| Weight decay | 0.0001 |
| Loss function | Binary Crossentropy |
| A hidden layer activation function | ReLU |
| Output layer Activation function | Softmax |
| Dropout | 0.5 |
| Momentum | 0.9 |

y-axis with a false positive rate close to 0 and a true positive rate close to 1. We obtain an AUC value of **98.65%** ascertaining the effectiveness of the model.

## 9.5. Evaluation of Space Complexity and Time Complexity based on down-scaling approach

This section evaluates the impact of down-scaling images in terms of time complexity and space complexity. The performance after applying bilateral and box blur methods is discussed as well to assess our down-scaling approach.

**9.5.1 Result analysis after applying Bilateral and Bilateral + Box blur filter.** To evaluate why two filters are employed simultaneously for down-scaling the images, a well comparison of accuracy, average image size and average training time is shown in Table 15.

It is observed from Table 15 that, in both cases, a test accuracy above 98% is obtained. The noticeable impact is, average image size can be reduced to 5–6 KB and training time from 110 seconds to 107 seconds when two filters are applied. Therefore, it asserts the effectiveness of applying both of these filters on the purpose of down-scaling without compromising accuracy.

**9.5.2. Evaluation of space complexity after applying down-scaling method.** The space complexity of ten randomly chosen preprocessed images (outputs of section 9.3) is analyzed based on their size. Down-scaling is applied to these images resulting in reduced storage requirements. The dataset generated by applying the down-scaling method is denoted as 'DG-after' and the dataset before applying down-scaling method is denoted as 'DG-before'.

In Fig 28, the orange bars show the storage requirements of the images which were not down-scaled whereas the blue bars show the storage requirements of the down-scaled images. It can be observed that for each image, approximately 40–60 KB storage was saved. As the storage requirements differ for different images, the total storage of 100 images before and after down-scaling is calculated. Before scaling down, the storage taken by 100 images was 180 Mb and after scaling down, the storage taken by 100 images was reduced to approximately 17 Mb.

**Table 15. Comparison among Bilateral and Bilateral + Box blur filter based on accuracy, average image size and average training time.**

| Experiment | Optimizer | Learning Rate | Average image size (KB) | Average training time (epoch/s) | Training Accuracy (%) | Validation Accuracy (%) | Test Accuracy (%) |
|---|---|---|---|---|---|---|---|
| Bilateral | Adam | 0.001 | 10–14 | 110 | 99.43 | 98.32 | 98.09 |
| Bilateral + Box blur | Adam | 0.001 | 5–6 | 107 | 99.66 | 98.49 | 98.74 |

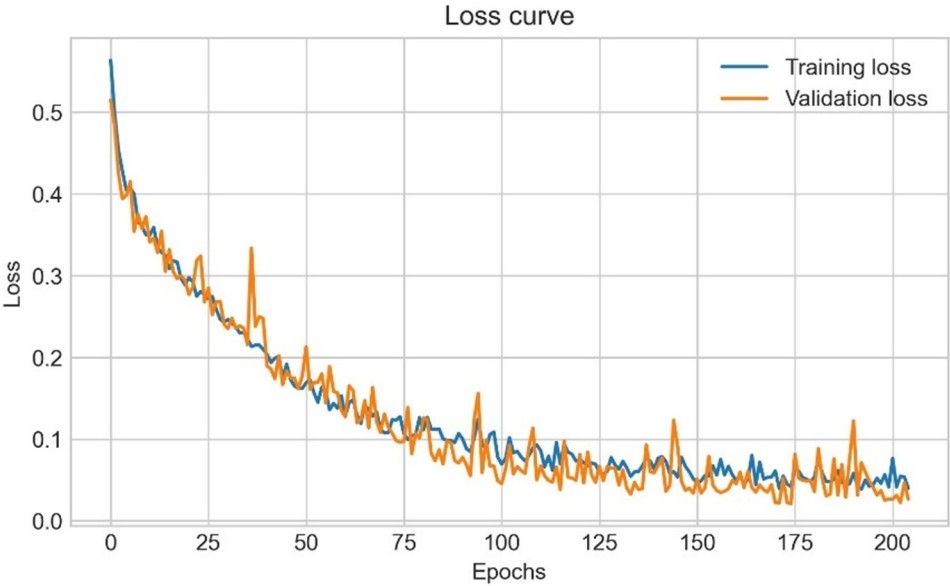

**Fig 24. Training and validation loss curve of 200 epochs for Adam with learning rate 0.001.**

**9.5.3. Evaluation of time complexity after applying down-scaling method.** As explained in section 9.2, the model is trained using optimizer Adam and learning rate 0.001 with both the preprocessed and the down-scaled dataset. Fig 29 shows the time taken for 200 training epochs for both of these datasets. Here, the x-axis represents the number of epochs and the y-axis represents the time taken by each epoch. The orange and blue line graphs visualize the training time of the datasets before (DG-before) and after (DG-after) applying down-scaling respectively. A significant gap can be seen between the orange and blue curves which demonstrate that the down-scaled dataset is consuming less training time.

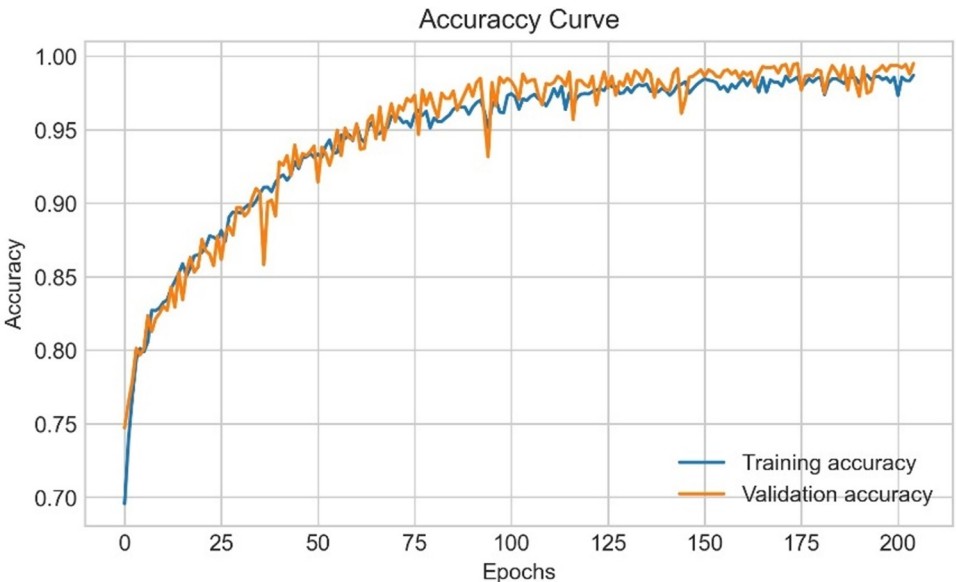

**Fig 25. Training and validation accuracy curve of 200 epochs for Adam with learning rate 0.001.**

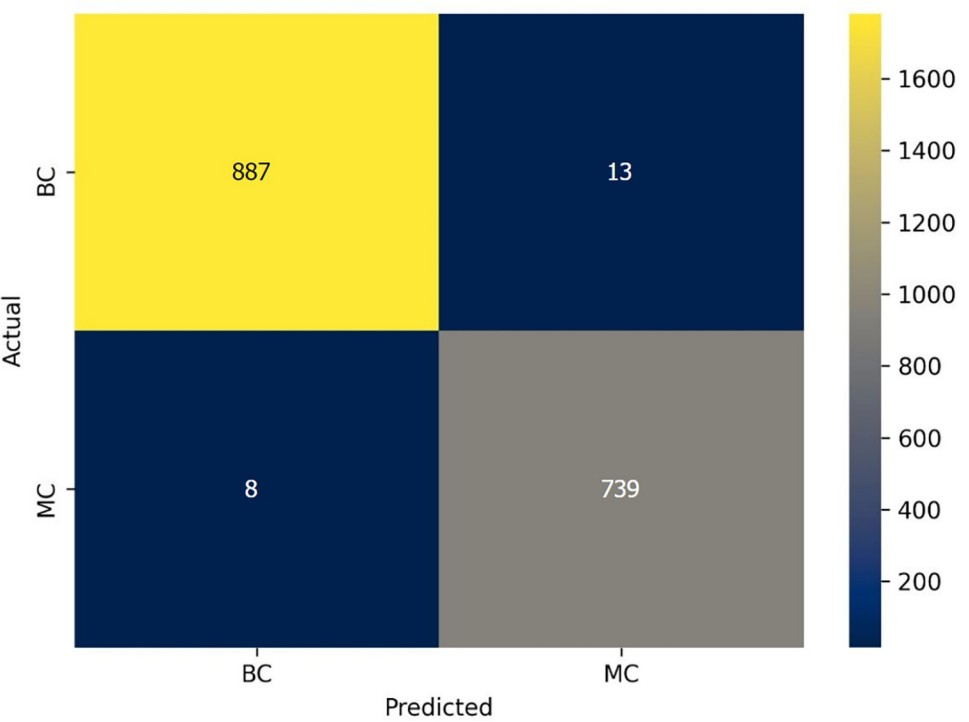

**Fig 26. Confusion matrix of the best configured model (Adam 0.001).**

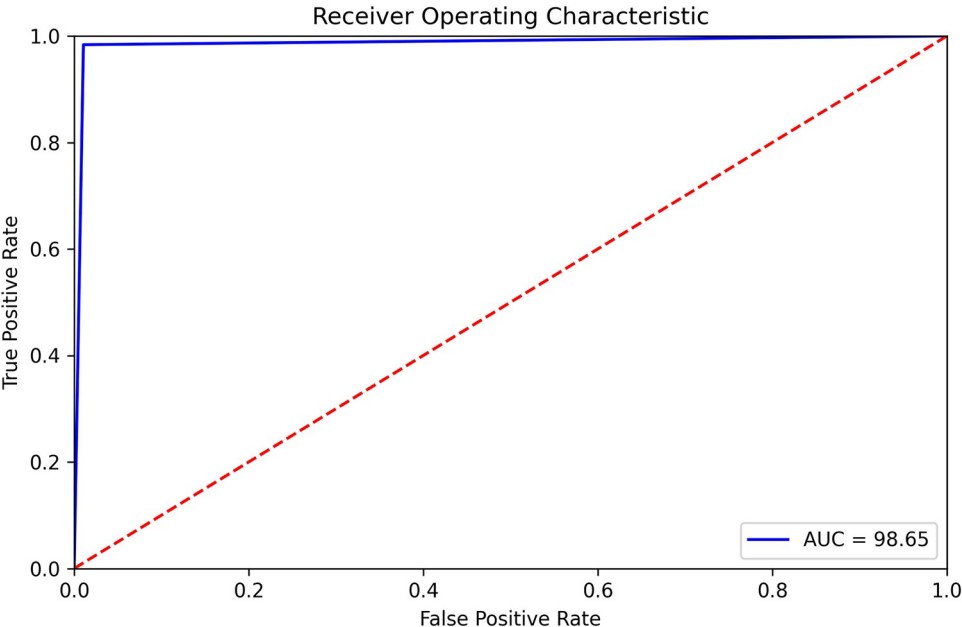

**Fig 27. ROC curve of best configured model (Adam 0.001).**

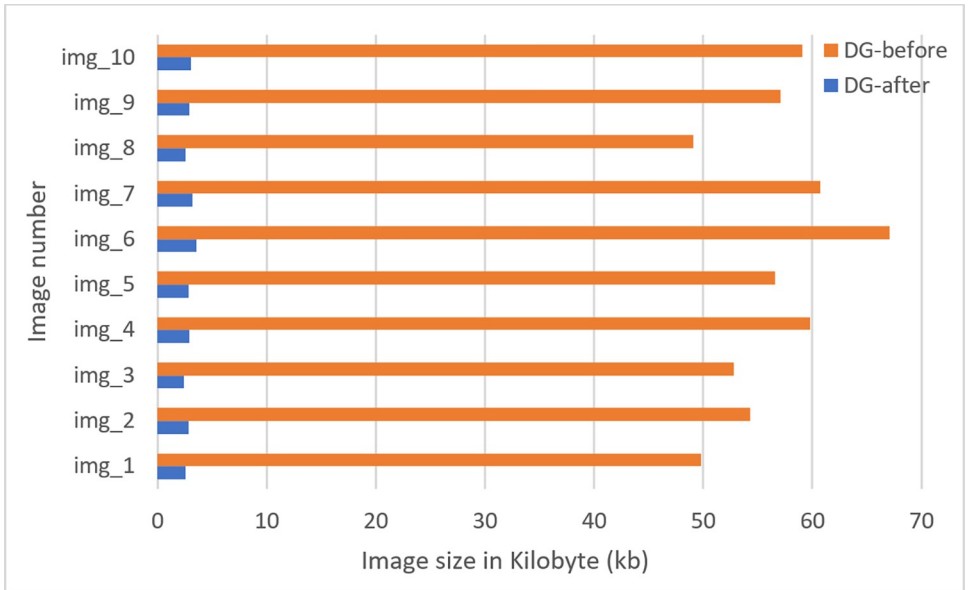

**Fig 28. Storage comparison for 10 images individually for both 'DG-before' and 'DG-after' where 'DG-before' indicates the dataset before applying down-scaling and 'DG-after' indicates the dataset generated after down-scaling.**

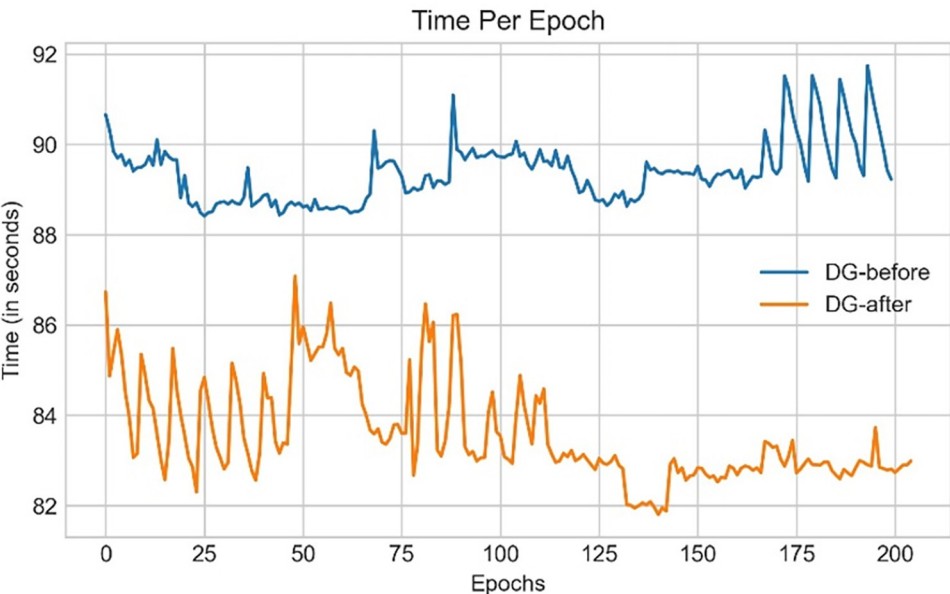

**Fig 29. Training time per epoch comparison for 200 epochs of datasets 'DG-before' and 'DG-after' where 'DG-before' indicates the dataset before applying downscaling and 'DG-after' indicates the dataset generated after down-scaling.** Another impact, observed during training is that our down-scaled dataset (DG-after) records the highest accuracy within 200 epochs (Fig 30) whereas the non-down-scaled dataset (DG-before) takes more than 200 epochs to achieve the highest accuracy. This further adds to the efficiency of employing this method on future researches.

**Table 16. Comparison of performance between geometric and photometric augmented dataset.**

| Experiment | T_Acc | T_Loss | Val_Acc | Val_Loss | Te_Acc | Te_Loss |
|---|---|---|---|---|---|---|
| Geometric | 96.29 | 0.18 | 97.54 | 0.27 | 97.03 | 0.23 |
| Photometric | 99.66 | 0.016 | 99.49 | 0.021 | 98.74 | 0.056 |

## 9.6. Performance analysis of geometric and photometric augmented dataset

To assess the performance of applying photometric augmentation technique on our dataset, some geometrical augmentation techniques, (vertical flip, horizontal flip, rotation by 90° and rotation by 45°) are explored. For both of the methods, our proposed model is trained using optimizer Adam with a learning rate of 0.001 and the experimental result is shown in Table 16.

In Table 16, training accuracy is denoted as T_Acc, training loss is denoted as T_Loss, Validation accuracy as Val_Acc, validation loss as Val_Loss, test accuracy as Te_Acc and test loss as Te_Loss. For geometric augmentation, poor performance is obtained compared to photometric augmentation. The test accuracy dropped at 97.03% from the highest accuracy of 98.74%. Though, the difference in the accuracies of these two data augmentation techniques is not significantly high for our dataset, while working with larger and real world skin cancer datasets, applying photometric augmentation might be a better approach.

## 9.7. Overfitting check employing k-fold cross-validation and splitting into various ratios

**9.7.1 K-fold cross-validation.** In this study, we have performed k-Fold cross-validation on the down-scaled dataset before and after applying augmentation, in order to evaluate proposed model's generalization capabilities as well as to check if there is any possibilities of overfitting issues. To check overfitting issue, K-fold cross-validation is performed for ensuring that every observation from the original dataset has the chance of appearing in training and test set which is a standard practice to detect overfitting in a CNN model [62]. For k-fold cross-validation, two k values of 5 and 10 are employed for experimentations as these are considered as standard k values [63]. Tables 17 and 18 represents the performance of 5-Fold and 10-Fold cross-validation respectively on augmented dataset.

In Tables 17 and 18, average validation accuracy is denoted as average val_acc. For 5-Fold cross-validation, the proposed model is able to achieve an average validation accuracy of 98.73% (Table 17). On the other hand, with 10-Fold cross-validation the proposed model gained an average validation accuracy of 98.31% (Table 18). Both in 5-fold and 10-fold cross-validation most of the folds had validation accuracies well above 96%. In both 5 and 10 fold cross-validation, no fold shows alarming gap between training accuracies and validation accuracies. Moreover, all the training and validation accuracies are quite balanced in every fold. This indicates no overfitting issues in our augmented dataset and adds to the robustness to proposed model.

For further validation, K-fold cross-validation is also carried out on the the dataset before applying augmentation with proposed model (Tables 19 and 20). Here the model is able to

**Table 17. Result of 5-Fold cross-validation on augmented dataset.**

| Accuracy | Fold 1 | Fold 2 | Fold 3 | Fold 4 | Fold 5 | Average Val_acc |
|---|---|---|---|---|---|---|
| Training | 98.42 | 99.64 | 99.73 | 98.62 | 99.57 | |
| Validation | 97.93 | 98.51 | 99.04 | 97.86 | 99.20 | 98.51 |

**Table 18. Result of 10-Fold cross-validation on augmented dataset.**

| Accuracy | Fold 1 | Fold 2 | Fold 3 | Fold 4 | Fold 5 | Fold 6 | Fold 7 | Fold 8 | Fold 9 | Fold 10 | Average Val_acc |
|---|---|---|---|---|---|---|---|---|---|---|---|
| Training | 97.34 | 99.86 | 96.40 | 99.88 | 99.51 | 99.74 | 98.86 | 99.07 | 98.47 | 99.61 | |
| Validation | 96.83 | 99.20 | 95.86 | 99.17 | 98.86 | 99.25 | 98.04 | 98.68 | 97.92 | 98.97 | 98.31 |

achieve an average validation accuracy of 98.29% and 98.26% relatively in 5 fold and 10 fold cross validation. In all folds the validation accuracies were well above 96%. Both 5 fold and 10 fold cross-validation, no fold shows alarming gap between validation and training accuracy. Furthermore, training and validation accuracies stays balanced in all folds indicating no over-fitting issues in non augmented dataset.

**9.7.2 Splitting the datasets before and after applying augmentation in different ratios.** We achieve the highest performance from the proposed model using a split ratio of training: validation: test = 70:20:10. To validate this perfoance further, by showing no occurance of overfitting, we have split the datasets before and after applying augmentation into three more ratios. The down-scaled dataset (before augmentation) is split into training:validation:test dataset using four ratios of 60:30:10, 75:15:10 and 90:10:10. Likewise, the same ratios are used to split the augmented dataset. Results of these experiments are showed in Tables 21 and 22.

It is observed that, for every ratio, the proposed model is able to acquire test accuracy close to our highest test accuracy on both augmented and non-augmented datasets. Moreover, the gap between training and test accuracy is minimal and quite balanced across all splitting ratios which indicate no overfitting in any splitting ratio and robustness of the model on any given similar dataset.

## 9.8. Performance analysis of the optimal model on noise induced test data

When working with real world datasets, in most cases, image quality got corrupted due to noise. As our proposed system provides an optimal accuracy on public dataset, it should be

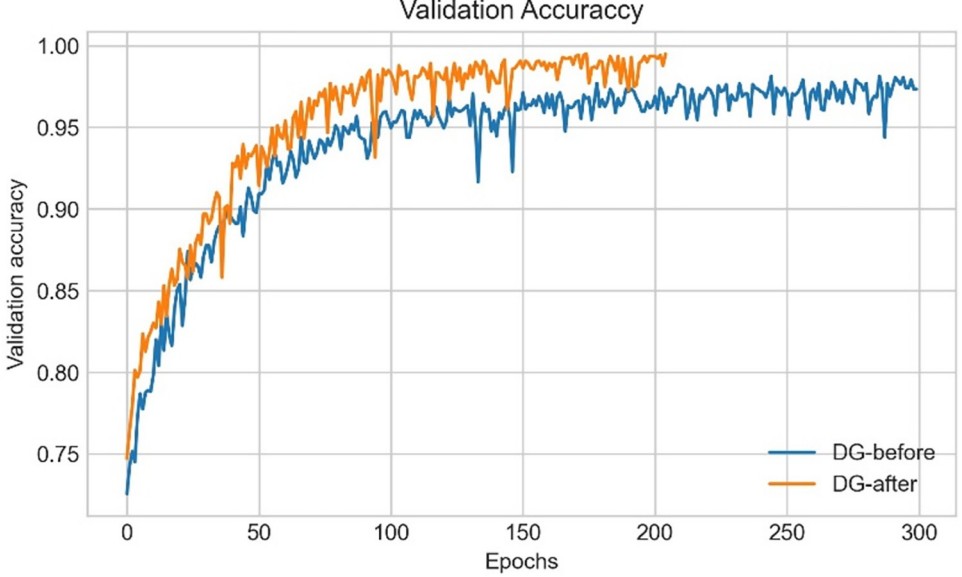

**Fig 30. Validation accuracy curve of both 'DG-before' and 'DG-after' datasets where 'DG-before' indicates the dataset before applying downscaling and 'DG-after' indicates the dataset generated after down-scaling.**

**Table 19. Result of 5-Fold cross-validation on the dataset before applying augmentation.**

| Accuracy | Fold 1 | Fold 2 | Fold 3 | Fold 4 | Fold 5 | Average Val_acc |
|---|---|---|---|---|---|---|
| Training | 98.81 | 99.36 | 98.52 | 99.37 | 98.54 | |
| Validation | 97.74 | 98.95 | 98.17 | 98.83 | 97.80 | 98.29 |

**Table 20. Result of 10-Fold cross-validation on the dataset before applying augmentation.**

| Accuracy | Fold 1 | Fold 2 | Fold 3 | Fold 4 | Fold 5 | Fold 6 | Fold 7 | Fold 8 | Fold 9 | Fold 10 | Average Val_acc |
|---|---|---|---|---|---|---|---|---|---|---|---|
| Training | 97.34 | 99.86 | 96.40 | 99.88 | 99.51 | 99.74 | 98.86 | 99.07 | 98.47 | 99.61 | |
| Validation | 97.58 | 98.42 | 99.02 | 96.95 | 98.37 | 97.86 | 98.75 | 99.10 | 98.42 | 98.21 | 98.26 |

**Table 21. Performance of proposed model on the dataset after augmentation with various splitting ratios.**

| Split ratio (Training:validation:test) | Optimizer | Learning Rate | Training Accuracy (%) | Validation Accuracy (%) | Test Accuracy (%) |
|---|---|---|---|---|---|
| 60:30:10 | Adam | 0.001 | 98.75 | 97.83 | 97.32 |
| 70:20:10 | Adam | 0.001 | 99.66 | 99.49 | 98.74 |
| 75:15:10 | Adam | 0.001 | 99.87 | 99.54 | 98.53 |
| 80:10:10 | Adam | 0.001 | 99.26 | 98.47 | 98.64 |

**Table 22. Performance of proposed model on the dataset before augmentation with various splitting ratios.**

| Split ratio (Training:validation:test) | Optmizer | Learning Rate | Training Accuracy (%) | Validation Accuracy (%) | Test Accuracy (%) |
|---|---|---|---|---|---|
| 60:30:10 | Adam | 0.001 | 98.54 | 97.71 | 96.83 |
| 70:20:10 | Adam | 0.001 | 99.46 | 98.73 | 98.67 |
| 75:15:10 | Adam | 0.001 | 99.71 | 98.28 | 98.62 |
| 80:10:10 | Adam | 0.001 | 99.34 | 98.63 | 98.51 |

examined how the network performs with noise induced test data set. Therefore, we have added Gaussian noise [64] with a value of 0.1 [65] and tested the model. After Gaussian noise was injected, the accuracy dropped a little but not significantly. It is observed that, before adding noise, we had the highest test accuracy of 98.7% and after adding noise, we get a test accuracy of 97.13% (Table 23). This outcome proves that even with noisy data, our proposed method is able to achieve optimal accuracy.

## 9.9. Comparison between the accuracy of the proposed methodology and existing classifiers

Ameri A. et al., [14] achieved an accuracy of approximately 84%, a sensitivity of 81%, a specificity of 88% and an AUC of 91% based on the Stochastic Gradient Descent with momentum (SGDM) optimizer, 40 epochs and batch sizes of 30 threshold confidence score of 0.5 and learning rate of 0.0001. Jinnai et. al., [19] obtained an accuracy of 91.5% on FRCNN model, using VGG-16 as backbone network, and SGD as optimizer with a learning rate of 0.0001. Other researchers [15] and [66] who used FGD optimizer, obtained lower accuracies of 94% and 95.2% based on the Faster-RCNN and F-MLP classifiers respectively. As described, Rehan

**Table 23. Performance of proposed model on noisy dataset.**

| noise | amount | Optimizer | Learning Rate | Test Accuracy (%) |
|---|---|---|---|---|
| Gaussian | 0.1 | Adam | 0.001 | 97.13 |

Ashraf et. al. [21] achieved an accuracy of 97.9% and 97.4% for their two datasets respectively. Lequan Yu et.al. [67] obtained an accuracy of 85.5% based on the FCRN model with SGD optimizer, with the ISBI 2016 Skin Lesion Analysis towards Melanoma Detection Challenge dataset. Though the number of epochs was more than 3000, only a batch size of 4 was considered for this experiment. An accuracy of 96% was achieved by Andre Esteva et. al. [68]. Their CNN model obtained this result based on the Stanford University Medical Center and undisclosed Online Databases holding 129,450 clinical images. An accuracy of approximately 92% accuracy was obtained using a CNN model in the research of Boman and Volminger [69] and Fujisawa et. al. [11].

Our approach is able to address the limitations of previous studies described in Table 1. All the image preprocessing algorithms and their parameter values deployed in this study are selected after a wide experiment with our dataset. Instead of applying geometrical augmentation, photometric augmentation is carried out to increase the number of images from 3297 to 16485 as photometric augmentation yields the highest performance. We propose a shallow CNN model named SCNN_12 by performing ablation study on it to acquire highest possible accuracy. The maximum accuracy **98.74**% is attained based on the Adam optimizer, batch size of 32, 200 epochs with a learning rate of 0.001. Table 24 shows a comparison of our model with existing work.

**Table 24. Comparison of accuracy between the proposed system and existing systems.**

| Authors | Models | Dataset | Batch Size | Epochs | Optimizer | Learning Rate | Accuracy |
|---|---|---|---|---|---|---|---|
| Ameri A. | CNN | HAM10000 dermoscopy image database (3400 images) | 30 | 40 | SGDM | 0.0001 | 84% |
| Lequan Yu et.al. | FCRN | ISBI 2016 Skin Lesion Analysis Towards Melanoma Detection Challenge | 4 | 3000 | SGD | 0.001 | 85.5% |
| Andre Esteva et. al. | CNN | The Stanford University Medical Center and undisclosed Online Databases (129,450 clinical images) | - - - - | - - - - | - - - - | - - - - | 96% |
| Shunichi Jinnai et. al. | FRCNN | 5846 clinical images from 3551 patients. | 4 | 100 | SGD | - - - - | 91.5% |
| Joakim Boman, et. al. | CNN | ISIC Dermoscopic Image Dataset (23,647), DermQuest (16,826), Dermatology Atlas (4,336), a total number of 1,948 images had been taken from DermAmin, Dermoscopy Atlas, Global Skin Atlas, Hellenic Dermatological Atlas, Medscape, Regional Derm, Skinsight and the pH2 database. | 100 | 30 | - - - - | 0.001 | 91% |
| Rehan Ashraf et. al. | CNN | DermIS: 3176 images<br>DermQuest: 1096 images | - - - - | 10 | - - - - | Small rate | 97.9%<br>97.4% |
| Manu Goyal et. al. | Faster-RCNN | International Symposium on Biomedical Imaging ISBI—2017 testing dataset | - - - - | - - - - | ——–– | - - - - | 94.5% |
| Abder-Rahman Ali et. al. | F-MLP | ISIC 2018: Skin Lesion Analysis Toward Melanoma Detection" grand challenge datasets | - - - - | 20 | FGD | 0.001 | 95.2% |
| Yasuhiro Fujisawa et. al. | CNN | ILSVR2012 dataset: Containing 1.2 million images within 1,000 classes | 30 | -–—–— | - - - - | - - - - | 92% |
| Proposed model SCNN_12 | **CNN** | **Kaggle skin cancer image ISIC archive dataset consisting of 3297 skin cancer images (1800 Benign and 1497 Malignant). After augmentation: 16485 images** | **64** | **200** | **Adam** | **0.001** | **98.74%** |

## 10. Limitation

We have worked with a small number of images. Though data augmentation technique enlarges the dataset, the performance of our proposed model could be more evaluated, experimenting with a larger dataset. Moreover, in some cases, real data differs from publicly available datasets. It could be investigated, how the model performs with real world data, if we could work with a real world dataset.

## 11. Conclusion

In this paper, we propose an automatic approach based on shallow CNN architecture to meet the challenges of skin cancer detection and classification in dermoscopy images. The approach is based on preprocessing the dermoscopy images using morphological closing and gamma correction, smoothing by bilateral filter, down-scaling applying box blur algorithm and augmenting the dataset using four augmentation techniques. We have also introduced segmentation where the ROI is extracted from the preprocessed images by applying morphological operations and thresholding. Compared with much deeper networks, the shallow CNN has a very good classification performance and a low computational time. Therefore, a shallow CNN network is developed and ablation study is employed to make it more robust. The system achieved an accuracy of 98.74% using the Adam optimizer with a learning rate of 0.001. In order to further validate this performance, the proposed model is experimented on noisy dataset and K-fold cross-validation as well as data splitting using various ratios is conducted. Therefore, our research demonstrates that lightweight CNNs with down-scaling mechanisms can be employed to solve complicated medical image analysis problems, reducing computational cost while preserving a high accuracy.

## Author Contributions

**Conceptualization:** Sidratul Montaha, Sayma Islam.

**Data curation:** Sidratul Montaha, A. K. M. Rakibul Haque Rafid.

**Formal analysis:** Sidratul Montaha, A. K. M. Rakibul Haque Rafid.

**Investigation:** Sidratul Montaha, Sami Azam, A. K. M. Rakibul Haque Rafid, Sayma Islam, Mirjam Jonkman.

**Methodology:** Sidratul Montaha, Sami Azam, A. K. M. Rakibul Haque Rafid, Sayma Islam, Pronab Ghosh.

**Supervision:** Sami Azam.

**Validation:** Pronab Ghosh.

**Visualization:** Sayma Islam, Pronab Ghosh.

**Writing – original draft:** Sidratul Montaha, A. K. M. Rakibul Haque Rafid, Sayma Islam, Pronab Ghosh.

**Writing – review & editing:** Sami Azam, Mirjam Jonkman.

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
