## [Decision Letter · Decision Letter 0]

15 Nov 2021

PONE-D-21-31272A shallow deep learning approach to classify skin cancer using down-scaling method to minimize time and space complexityPLOS ONE

Dear Dr. Azam,

Thank you for submitting your manuscript to PLOS ONE. After careful consideration, we feel that it has merit but does not fully meet PLOS ONE’s publication criteria as it currently stands. Therefore, we invite you to submit a revised version of the manuscript that addresses the points raised during the review process.

 Please revise your paper according to the comments from reviewer 1 (especially on the explanation on the selected parameters and hyperparameters values as well as adding noise to the current data to test the robustness of your model, and also explain in detailed the limitation of your model or research), and reviewer 2 (especially on suggesting you to perform a k-fold cross validation on their data as well as performing the data splits with various ratios and some data splits before doing augmentation).

We look forward to receiving your revised manuscript.

Kind regards,

Yan Chai Hum

Academic Editor

PLOS ONE

Additional Editor Comments (if provided):

Please refer to reviewers' comments

3. PLOS requires an ORCID iD for the corresponding author in Editorial Manager on papers submitted after December 6th, 2016. Please ensure that you have an ORCID iD and that it is validated in Editorial Manager. To do this, go to ‘Update my Information’ (in the upper left-hand corner of the main menu), and click on the Fetch/Validate link next to the ORCID field. This will take you to the ORCID site and allow you to create a new iD or authenticate a pre-existing iD in Editorial Manager. Please see the following video for instructions on linking an ORCID iD to your Editorial Manager account: https://www.youtube.com/watch?v=_xcclfuvtxQ. 

4. Please ensure that you refer to Figures 1, 2, 8 in your text as, if accepted, production will need this reference to link the reader to the figure.

Reviewers' comments:

Reviewer's Responses to Questions

**Comments to the Author**

1. Is the manuscript technically sound, and do the data support the conclusions?

Reviewer #1: Yes

Reviewer #2: Partly

2. Has the statistical analysis been performed appropriately and rigorously? 

Reviewer #1: Yes

Reviewer #2: Yes

3. Have the authors made all data underlying the findings in their manuscript fully available?

Reviewer #1: Yes

Reviewer #2: Yes

4. Is the manuscript presented in an intelligible fashion and written in standard English?

Reviewer #1: Yes

Reviewer #2: Yes

5. Review Comments to the Author

Reviewer #1: This research classifies skin cancer lesions as benign or malignant. For the classification of lesions into benign and malignant, the authors suggest using a deep learning model with a shallow architecture to categorize the lesions. It is trained and tested using Kaggle skin cancer data by applying various augmentation methods, and the model achieves an accuracy of 98.87 percent, according to the results of the evaluation. This study confirms that successful training for medical image analysis may be achieved by addressing the issues of training time and spatial complexity.

The research topic addressed by this work is both timely and valuable. Overall, I found this research to be extremely fascinating, however there are several problems that need to be addressed. I'm sure the authors could easily address each of the problems raised. I believe the research will make a valuable addition to the discipline. The following suggestions are made in order to enhance the overall quality of the paper:

1. The introduction section is written well, only the author needs to mention the structure of the paper at the end of this section.

2. Literature Review: Description of related works presented in the Related Works section should be written as a critic point of view. Pros and cons of these methods and how the proposed method has overcome the drawbacks of these methods that also should be explained. Use table to summarizes the literature in terms of the task and the techniques that were employed, include a column that highlights the limitations of the-state-of-the-art approaches.

3. Figures: Label the x and y axes in the figures: Figure 23 and Figure 25.

4. Algorithm: In All the algorithms you must define Input and Output which are missing.

5. How the values of parameters and hyper-parameters used in the experiment are determined that needs more discussions.

6. In order to assert the Robustness of their model, the authors must show model performance as noise rises, i.e., as we move farther away from real data. How they simulate situations where the image quality is corrupted? I suggest that you add some more results on evaluation on noisy data.

7. What is the limitation of this study?

8. You have used many variables, abbreviations, Nomenclature in the article. From this perspective, an Index of Notations and Abbreviations would be beneficial for a better understanding of the proposed work. Furthermore, please check carefully if all the abbreviations and notations considered in the work are explained for the first time when they are used, even if these are considered trivial by the authors.

Reviewer #2: This paper presents a deep learning approach for classifying skin cancer that demonstrates a very high accuracy score while minimizing time and space complexity. In my view the paper may have important findings and potentially introducing a method that may be a step up with respect to the state of the art. However, in its current form there are numerous issues, which warrant a major revision of this study before it can be considered for publication.

First, I would like to give a summary of these issues, then for each of them I will go into specific details:

(1) Main findings of the paper: A very high accuracy and F1 score is achieved by the introduced method. This should always raise the possibility of potential overfitting, and based on the presented data validation against overfitting is in my view not sufficient in the paper. That is, currently it seems questionable whether the displayed scores do tell the real performance of the presented method.

(2) Presentation of the paper and technical description: the study is quite lengthy and while for certain topics the authors strive to put all tiny details, this becomes even rather superfluous in some cases, while on the other hand some really important parts/data are missing or too vague. There are also some unnecessary repetitions in the paper, and all in all, in certain aspects these issues make the presentation somewhat convoluted and also unclear on some points.

Now, let's dwell deeper into these main issues.

Regarding (1), immediately it has to be noted that the initial number of images are not high in the context of training a neural network, which is understandable though because of the relative scarce availability of big skin cancer image databases. However, a further issue can be that usually the inherent diversity of dermoscopic images is not necessarily high. In this way one could expect that a study introducing a new neural network for detecting malign skin lesions would put a sufficiently strong emphasis on augmentation techniques as well as on validation of accuracy scores. Unfortunately, while the paper goes to great length on defining some of the pre-processing techniques applied to the images, when it comes to augmentation, the authors only state they are carrying out contrast and brightness alteration to the original set. It is not even defined, what exactly these transformations are, while justification for not using other common techniques in addition is not made, either. (see section 8.1, lines 649-656). In my opinion this part of the study needs a much greater elaboration. Please insert the math definitions of brightness and contrast transformation and what parameters values were exactly used, and also explore the additional augmentation possibilites.

Furthermore, only a single data split was used throughout the paper, and this split has been done after carrying out augmentation (section 8.2). In this way, it's hard to judge whether the images in the training set sufficiently differ from the ones in the validation and testing sets. In my view, although Fig. 27. exhibiting the training and validation loss curves would suggest that no overfitting occurs, a more rigorous check has to be carried out to see if this is indeed true. I suggest the authors perform a k-fold cross validation on their data as well as performing the data splits with various ratios and some data splits before doing augmentation. It must be demonstrated that the achieved very high accuracy scores hold up against such shuffling and cross validation.

As for technical description and presentation of the paper, there are numerous issues that would need to be corrected/justified:

- Section 7.1 (Image Pre-Processing): The authors describe various possible algorithms for hair removal and enhancement. They state the algorithms producing the highest PSNR are finally selected to be used in their method. Table 1 displays the measured PSNR values. However, despite that gamma correction provides the second worst value from 4 algorithms for enhancement, finally it has been selected for use as the enhancement algorithm. This is in clear contradiction with the authors' selection principle. Another issue: throughout the paper the various algorithms are described with pseudo-code. This would be fine, but already Algorithm 1 in Section 7.1 is a source of confusion, since it's not an actually used algorithm within the processing method, but rather the selection process for the hair removal and enhancement algorithms. I suggest to remove this, and only provide pseudo-code for the actual image processing subalgorithms themselves.

- Line 315, formula (1): Only dilation is defined, the other formulae are missing (compare them with those displayed in Ref. [25])

- Section 7.1.2.1. Gamma Correction: the authors declare that they have used a gamma value of 1.2, "as higher gamma values were washing out the pixels a lower gamma values caused a loss of important details.". My problem with this is that visually this might be true, but did they check if other gamma values were causing also worse performance of their neural network? The other issue is that images taken with various cameras and in different exposure/illumination settings, but even the used color space in the recorded images can also highly influence what gamma correction value would look optimal. Therefore, 1.2 might be suitable for the given set of images the authors used, but it may be completely suboptimal for an other image set from a different source. Overall, since various gamma correction values can cause drastically different output, sensitivity of the applied neural network in terms of accuracy must be checked against this parameter. Could you please do such check or in any way justify why a universal value of 1.2 would be suitable for any image sets in which skin cancer lesions should be detected?

- Lines 391-393, formulae (4-6): please define what the subindices of 0, 1, 2 for various parameters (, ) are, and complete the definition of all parameters in the succeeding text part

- Line 412, formula (7): this is repetition of formula (1). see my comments above: complete formula (1) and refer back to that one in this subsection

- Lines 416-417, formula (8): please define what is F and S

- Line 475, formula (9): what indices k, k-1 signify? define.

- Section 7.3. Down-sampling: this section would need some revision. First, the authors state that they are down-sampling the images. However, what is really happening here is an edge-preserving low-pass filtering by using a bilateral filter, followed by a (non-edge preserving) low-pass filtering by a box filter. It is not down-sampling per se, unless the result of the filtering is truly downsampled. I presume that the original resolution of 224x224 pixels is reduced to 112x112, given the radius of the box filter, but it is nowhere stated in the text explicitly. Could you please clarify? The second issue: it's not completely understandable why both of these filters are needed, if finally the edge-preserving nature of bilateral filtering gets lost anyway. In my view one filter would be enough, it can be bilateral filtering itself with well selected parameters. Third issue: the authors do not define, what parameter values are exactly used for bilateral filtering ( and ), which can highly influence the output. Finally, I see the lengthy definition of box filtering in 7.3.2 superfluous, as it's a very basic filter well known in the literature. The subsection could be shortened significantly.

- Lines 578-580, formula (14): the definition of parameters is inconsistent and incomplete: the text refers to O denoting the ground truth image, while the formula has G; m and n are not defined.

- Line 603, formula (17): what is d_fi and d_d? please define.

- Section 8.1, Augmentation: see my comments above, this section would need a serious rework, brightness and contrast enhancement algorithms must be mathematically defined and the exact parameter values used for these displayed.

- Line 743, formula (22): again, parameters are not defined in the text, please complete.

- Line 865, formula (32): this is repetition of formula (17). Please delete this one and refer back to the previous definition.

- Section 10.6. This is in great part a repetition of section 4 (Literature review). Please compact one of these to avoid unnecessary duplication, I would say preferably Section 10.6. should be the more detailed one, as here the direct comparison is made with the newly introduced method.

Issues with figures:

- Fig.3.: it is a rather convoluted image, and together with Fig. 4, with which there is a significant overlap, it is hard to follow, what is happening here. I would suggest a clean-up and possibly a well visible dissecting of image processing steps from all else, so that contents of Fig. 3 and Fig. 4. be set apart.

- Fig. 9.: this flow chart mixes the processing step of gamma correction with finding a good gamma value (that is finally set to 1.2 for all images any way). There is no need for this, as it adds to the confusion of what is the actual processing and what is selection process for good algorithms and their parameters. Simplify the figure to simply display the gamma correction itself.

- Fig. 19: I see this figure superfluous, it can be deleted.

- Fig. 22: what do the numbers 34567 (number of images??), and 56789 (on top of data augmentation??) signify? These must be incorrect values in my view, please check and/or explain.

- Fig.31. figure legend is missing for the displayed numbers (file sizes in kBytes?)

- Fig.32 and Fig.34: they are basically showing pairs of numbers compared (those of drive space and training time), which is really not necessary. I suggest to delete these figures.

Other minor issues:

Please carry out a thorough proof reading. There are numerous typographical errors or styling issues in these lines: 84, 123, 127, 147, 214, 309, 340, 354, 367, 375, 388, 586, 594-596, 642, 759, 842, 868. These include improper styling of mathematical parameters, that have subindices but these are not put into superscript or subscript format.

Reference 11: authors are missing

Reference 41: what is this? It is not intelligible in this form.

6. PLOS authors have the option to publish the peer review history of their article (what does this mean?). If published, this will include your full peer review and any attached files.

Reviewer #1: No

Reviewer #2: No

---

## [Author Response · Author response to Decision Letter 0]

24 Feb 2022

All the comments from the reviewers and the editor are addressed and made necessary changes to the paper. The paper has been revised according to the comments from reviewer 1 (especially on the explanation on the selected parameters and hyperparameters values as well as adding noise to the current data to test the robustness of your model, and also explain in detailed the limitation of your model or research), and reviewer 2 (especially on suggesting you to perform a k-fold cross validation on their data as well as performing the data splits with various ratios and some data splits before doing augmentation).

The followings files are uploaded:

All the necessary files and figures are attached with this submission. All the following items are included:

---

## [Decision Letter · Decision Letter 1]

23 May 2022

PONE-D-21-31272R1A shallow deep learning approach to classify skin cancer using down-scaling method to minimize time and space complexityPLOS ONE

Dear Dr. Azam,

Thank you for submitting your manuscript to PLOS ONE. After careful consideration, we feel that it has merit but does not fully meet PLOS ONE’s publication criteria as it currently stands. Therefore, we invite you to submit a revised version of the manuscript that addresses the points raised during the review process.

Please revise the article in accordance to reviewer's comment regarding the PSNR.

We look forward to receiving your revised manuscript.

Kind regards,

Yan Chai Hum

Academic Editor

PLOS ONE

Journal Requirements:

Additional Editor Comments:

Please consider to revise according to reviewer's comments.

Reviewers' comments:

Reviewer's Responses to Questions

**Comments to the Author**

1. If the authors have adequately addressed your comments raised in a previous round of review and you feel that this manuscript is now acceptable for publication, you may indicate that here to bypass the “Comments to the Author” section, enter your conflict of interest statement in the “Confidential to Editor” section, and submit your "Accept" recommendation.

Reviewer #1: All comments have been addressed

Reviewer #2: All comments have been addressed

2. Is the manuscript technically sound, and do the data support the conclusions?

Reviewer #1: Yes

Reviewer #2: Yes

3. Has the statistical analysis been performed appropriately and rigorously? 

Reviewer #1: Yes

Reviewer #2: Yes

4. Have the authors made all data underlying the findings in their manuscript fully available?

Reviewer #1: Yes

Reviewer #2: Yes

5. Is the manuscript presented in an intelligible fashion and written in standard English?

Reviewer #1: Yes

Reviewer #2: Yes

6. Review Comments to the Author

Reviewer #1: In the revised version, the authors have addressed all of my concerns, and the article is a lot better than before. I recommend to accept this article.

Reviewer #2: The authors clearly addressed each of the raised questions, concerns and they have carried out a serious rework on the manuscript. I only found one minor outstanding issue that needs to be corrected, after which I see that the paper can be accepted for publishing. This is concerned around the question of choosing 1.2 as the optimal gamma value when applying gamma correction as pre-processing on the images.

The authors responded to my concern by providing MSE, PSNR and similar metrics for various gamma values, comprised in Table 4 of the revised manuscript. In my opinion this does not provide a proper justification, and the values presented in this table do not make much sense, since it's trivial that 1.2 is the closest gamma value to 1 (that gives an identity transformation) from among the other listed candidates, and as such it is to be expected that it will yield the highest PSNR compared to that when applying the other gamma values. However, if the authors applied a gamma value of 1.05, that would generate an even higher PSNR, as it's even closer to 1.

Therefore, please delete Table 4 and the corresponding text on PSNR values (lines 465-467). It remains acceptable, though, that heuristically a gamma value of 1.2 provided the best performance of the applied model, together with the caveat, that it might not be a universally applicable value to all datasets outside of the ones used in this study.

7. PLOS authors have the option to publish the peer review history of their article (what does this mean?). If published, this will include your full peer review and any attached files.

Reviewer #1: No

Reviewer #2: No

---

## [Author Response · Author response to Decision Letter 1]

25 May 2022

Dear Editor,

Thank you for allowing a resubmission of our manuscript, with an opportunity to address the reviewers’ comments. There is one minor concern raised by reviewer 2 which we have addressed in our current submission.

We are uploading (a) our point-by-point response to the comments (below) (response to reviewers), (b) an updated manuscript with yellow highlighting indicating changes, and (c) a clean updated manuscript without highlights.

We look forward to hearing from you and respond to any further queries you may have.

Best regards,

Sami Azam et al.

Reviewer #2, concern #1: 

Concerns: The authors responded to my concern by providing MSE, PSNR and similar metrics for various gamma values, comprised in Table 4 of the revised manuscript. In my opinion this does not provide a proper justification, and the values presented in this table do not make much sense, since it's trivial that 1.2 is the closest gamma value to 1 (that gives an identity transformation) from among the other listed candidates, and as such it is to be expected that it will yield the highest PSNR compared to that when applying the other gamma values. However, if the authors applied a gamma value of 1.05, that would generate an even higher PSNR, as it's even closer to 1.

Therefore, please delete Table 4 and the corresponding text on PSNR values (lines 465-467). It remains acceptable, though, that heuristically a gamma value of 1.2 provided the best performance of the applied model, together with the caveat, that it might not be a universally applicable value to all datasets outside of the ones used in this study.

Author action: Table 4 and the corresponding text on PSNR values (lines 465-467) have been deleted.

---

## [Editor Report · Decision Letter 2]

31 May 2022

A shallow deep learning approach to classify skin cancer using down-scaling method to minimize time and space complexity

PONE-D-21-31272R2

Dear Dr. Azam,

We’re pleased to inform you that your manuscript has been judged scientifically suitable for publication and will be formally accepted for publication once it meets all outstanding technical requirements.

Kind regards,

Yan Chai Hum

Academic Editor

PLOS ONE
---

## [Editor Report · Acceptance letter]

6 Jun 2022

PONE-D-21-31272R2 

A shallow deep learning approach to classify skin cancer using down-scaling method to minimize time and space complexity 

Dear Dr. Azam:

I'm pleased to inform you that your manuscript has been deemed suitable for publication in PLOS ONE. Congratulations! Your manuscript is now with our production department. 

Kind regards, 

on behalf of

Dr. Yan Chai Hum 

Academic Editor

PLOS ONE